# FIND: A Function Description Benchmark for Evaluating Interpretability Methods

**Sarah Schwettmann**[1]* **Tamar Rott Shaham**[1]*
**Joanna Materzynska**[1] **Neil Chowdhury**[1] **Shuang Li**[1]
**Jacob Andreas**[1] **David Bau**[2] **Antonio Torralba**[1]

[1]MIT CSAIL    [2]Northeastern University

## Abstract

Labeling neural network submodules with human-legible descriptions is useful for many downstream tasks: such descriptions can surface failures, guide interventions, and perhaps even explain important model behaviors. To date, most mechanistic descriptions of trained networks have involved small models, narrowly delimited phenomena, and large amounts of human labor. Labeling all human-interpretable sub-computations in models of increasing size and complexity will almost certainly require tools that can generate and validate descriptions automatically. Recently, techniques that use learned models in-the-loop for labeling have begun to gain traction, but methods for evaluating their efficacy are limited and ad-hoc. How should we validate and compare open-ended labeling tools? This paper introduces FIND (**F**unction **IN**terpretation and **D**escription), a benchmark suite for evaluating the building blocks of automated interpretability methods. FIND contains functions that resemble components of trained neural networks, and accompanying descriptions of the kind we seek to generate. The functions are procedurally constructed across textual and numeric domains, and involve a range of real-world complexities, including noise, composition, approximation, and bias. We evaluate methods that use pretrained language models (LMs) to produce code-based and natural language descriptions of function behavior. Additionally, we introduce a new interactive method in which an Automated Interpretability Agent (AIA) generates function descriptions. We find that an AIA, built with an off-the-shelf LM augmented with black-box access to functions, can sometimes infer function structure—acting as a scientist by forming hypotheses, proposing experiments, and updating descriptions in light of new data. However, FIND also reveals that LM-based descriptions capture global function behavior while missing local details. These results suggest that FIND will be useful for characterizing the performance of more sophisticated interpretability methods before they are applied to real-world models.

## 1 Introduction

The central task of interpretability research is to explain the functions that AI systems learn from data. Investigating these functions requires experimentation with trained models, using tools that incorporate varying degrees of human input. Hand-tooled approaches that rely on close manual inspection [Zeiler and Fergus, 2014, Zhou et al., 2014, Mahendran and Vedaldi, 2015, Olah et al., 2017, 2020, Elhage et al., 2021] or search for predefined phenomena [Wang et al., 2022, Nanda et al., 2022] are increasingly complemented by more automatic approaches that enable larger-scale analysis [Bau et al., 2020, Mu and Andreas, 2020, Hernandez et al., 2022, Oikarinen and Weng, 2023,

---

*Indicates equal contribution. Address correspondence to: schwett@mit.edu, tamarott@mit.edu

37th Conference on Neural Information Processing Systems (NeurIPS 2023) Track on Datasets and Benchmarks.

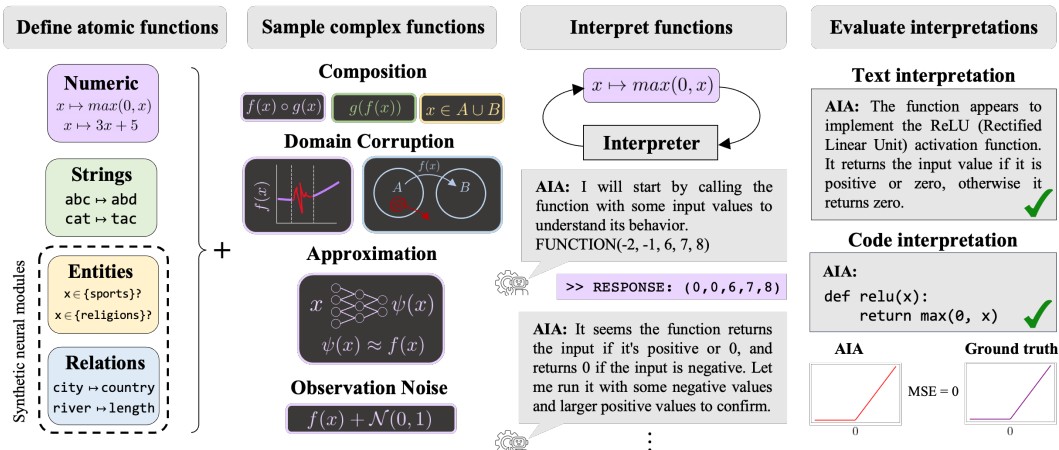

**Figure 1: The FIND benchmark.** FIND is constructed procedurally: atomic functions are defined across domains including elementary numeric operations (purple), string operations (green), and synthetic neural modules that compute semantic similarity to reference entities (yellow) and implement real-world factual associations (blue). Complexity is introduced through composition, bias, approximation and noise. We provide an LM-based interpretation baseline that compares text and code interpretations to ground-truth function implementations.

Conmy et al., 2023]. Selection between approaches is highly application-dependent; to date, no single protocol answers all queries users might have about a system [Doshi-Velez and Kim, 2017, Vaughan and Wallach, 2020]. However, considering the growing body of evidence that LMs are capable of complex reasoning and problem-solving across domains [Wei et al., 2022, OpenAI, 2023, Yao et al., 2023, Lightman et al., 2023], we recognize their potential to become backbones of generalized agents for automated interpretability. Indeed, recent open-ended techniques have used pretrained LMs to describe the behavior of black-box text modules [Singh et al., 2023], including individual units inside other LMs [Bills et al., 2023]. As we enter a regime where model explanation is performed by models that are themselves uninterpretable, external evaluations of these techniques will be vital. At present, evaluation of network description procedures is limited and bespoke, in part because measuring performance on real-world problems is difficult when ground-truth descriptions of network structure are unknown [Doshi-Velez and Kim, 2017, Lipton, 2018, Miller, 2019, Hooker et al., 2019].

This paper introduces **FIND** (**F**unction **IN**terpretation and **D**escription), a benchmark suite for evaluating the building blocks of automated interpretability methods on functions whose structure is known *a priori* (see Figure 1). FIND is built from over 2000 procedurally generated function interpretation problems (*e.g.* $f$ : country $\mapsto$ capital, unless country is in South America, in which case $f$ returns undefined). In each problem, candidate interpretation procedures (interpreters) are given black-box access to functions, optionally accompanied by metadata (*e.g.* the domain of the function). After evaluating these functions on chosen inputs (*e.g.* Japan, Mexico, Peru), interpreters must eventually return a structured description that can be evaluated against other descriptions or used to simulate function behavior.

We also introduce a new interpretation method that uses Automated Interpretability Agents (AIAs) to interactively probe functions and explain their behavior. We formulate our approach to the interpretation problem as an implementation of the scientific method, wherein the goal of the AIA is to describe the process underlying observed input–output relations. In contrast to existing full-text explanation systems that apply automatic captioning methods (e.g. [Hernandez et al., 2022, Bills et al., 2023]) to pre-selected input–output pairs, the AIA generates the data itself, by running the function on inputs it selects, observing the outputs, and updating hypotheses until it can describe the function to human end-users. We evaluate both AIAs and existing, non-interactive automated interpretability methods on **FIND**. While exhibiting sophisticated experimentation strategies and out-performing non-interactive methods, the top-performing AIA nonetheless fails to adequately describe 48% of functions. **FIND** thus shows that, in spite of promising recent results, methods based on current LMs alone are unlikely to robustly automate even high-level interpretability tasks.

**FIND** focuses on the black-box function description paradigm because black-box description appears as a subroutine (or is the sole operation) implemented by almost every existing automated interpretation method, spanning label retrieval, program synthesis, and learning-based approaches. The

functions that make up **FIND** are derived from past findings about specific phenomena observed in the wild, including polysemanticity [Olah et al., 2020, Gurnee et al., 2023], compositionality [Fong and Vedaldi, 2018, Mu and Andreas, 2020], and task misspecification [Grünwald and Van Ommen, 2017], and thus exercise the kinds of interpretability behaviors that will be required for understanding real neural networks. While not the focus of our initial release, **FIND** is extensible, and designed to eventually support targeted evaluation of `interpreters'` ability to identify disparate model accuracy across input regions (relevant to fairness and accountability) and shortcut solutions to complex algorithmic problems (relevant to reasoning and robustness). We intend for it to be a living benchmark, incorporating new functions, interaction paradigms, and evaluation metrics, as interpretability methods grow in sophistication and improve our understanding of real-world model behavior.

## 2 FIND functions

**FIND** is built from a diverse set of programs including explicit mathematical equations, string operations, logical composition, lexical relations, and factual associations. Our motivation is to construct a set of tasks that assess how well an automated `interpreter` formulates and tests hypotheses about an opaque system, and based on the results of those experiments, writes a language description or a program approximating the system. Using additional automated evaluation procedures that we describe in Section 3.1, language descriptions are compared to ground-truth function explanations, and code-based descriptions are evaluated for how accurately they reproduce function behavior.

Each function in **FIND** is defined inside an independent Python script that accepts input arguments from the `interpreter` and returns outputs to the command line. We construct **FIND** by defining a set of atomic tasks in different domains and a set of operators that introduce additional complexity. Numeric functions (Section 2.1) and string functions (Section 2.2) are expressed under a common API that supports sampling parameters and composing, corrupting, and approximating functions. For functions involving lexical semantics and factual associations, we implement synthetic neural modules with an LM backbone (Section 2.3). Table 1 shows example functions from each category.

The **FIND** API and benchmark have been open-sourced under the MIT License and are available at: https://github.com/multimodal-interpretability/FIND.

### 2.1 Functions with numeric inputs

Numeric functions test a variety of capabilities needed to interpret basic mathematical operations in neural networks, which can include relations between neurons or a model's representation of numeric

**Table 1:** Example **FIND** functions.

| Category | Example Atomic Functions | Example Complex Functions |
|---|---|---|
| Numeric Functions | `linear` $x \mapsto ax + b$ 
 `periodic` $x \mapsto \sin\left(\frac{2\pi}{b}(x - c)\right)$ 
 `absolute` $x \mapsto |x|$ 
 `relu` $x \mapsto \max(x, 0)$ 
 `sqrt` $x \mapsto \sqrt{x}$ 
 `constant` $x \mapsto a$ 
 `rational` $x \mapsto \frac{x}{x+a}$ 
 `reciprocal` $x \mapsto 1/x$ | *Composition* 
 `linear` ∘ `absolute` $x \mapsto (ax + b) + |x|$ 

 *Domain corruption* 
 `corruption` $x \mapsto |x|$ if $x \in [a, b]$ else $\epsilon$ 

 *Observation noise* 
 `linear + noise` $x \mapsto ax + b + \epsilon$ |
| String Functions | `capitalize` '*apple*' ↦ APPLE 
 `reverse` '*apple*' ↦ '*elppa*' 
 `replace(a,b)` '*apple*' ↦ '*bpple*' 
 `shift_last` '*apple*' ↦ '*appld*' | *Composition* 
 `replace(reverse)` '*apple*' ↦ '*elppb*' 
 `reverse(shift_last)` '*apple*' ↦ '*flppa*' 
 `capitalize(replace)` '*apple*' ↦ BPPLE |
| Neural Modules | *Entities* 
 time and duration 
 winter sports 
 plants and botany 
 *Relations* 
 country → capital city 
 gemstone → color | *Composition* 
 time and duration `OR` winter sports 
 winter sports `OR` plants and botany 
 time and duration `OR` plants and botany 
 *Domain corruption* 
 country → capital `EXCEPT` in Asia (return 0) 
 gemstone → color `EXCEPT` if red (return 0) |

inputs. One example is Nanda et al. [2022], where an algorithm using trigonometric manipulations was found to perform modular arithmetic in a toy transformer model. We sample 1000 numeric functions from the **FIND** API for the benchmark dataset. $85\%$ are parameterized atomic functions under noiseless and noisy conditions, and $15\%$ are compositions.

**Atomic functions** are defined as explicit functions found in many mathematical and scientific computing libraries such as Python [Van Rossum, 2020], SciPy [Virtanen et al., 2020], and NumPY [Harris et al., 2020], as well as standard neural activation functions such as ReLU [Fukushima, 1975, Nair and Hinton, 2010]. Table 1 shows examples of atomic functions defined in **FIND**. For each function in the set of atomic functions $\mathcal{A}$, we sample native parameters, scaling factor $a$, and bias $b$. Parameters and sampling procedure details are provided in the Appendix and the **FIND** API.

**Composition** of atomic functions $f(x)$ and $g(x)$ applies an operator sampled from $C = \{\cdot, +\}$, where $f \circ g = f(x) \cdot g(x)$ or $f(x) + g(x)$. Composed functions are sampled from a subset $\mathcal{A}_C$ of atomic functions $\mathcal{A}$ to limit final complexity. $f(x), g(x) \in \mathcal{A}_C$ are described in the Appendix.

**Observation noise** added to $f(x)$ tests how well the interpreter is able to estimate an underlying function in the presence of additive noise, and whether it is able to distinguish between different types of noise. $15\%$ of functions $f(x) \in \mathcal{A}$ in the **FIND** benchmark are sampled with additive noise $f(x) + X$, where $X$ follows either a Normal, Uniform, or Poisson distribution.

**Domain corruption** replaces function values locally with random values. This is done either inside or outside of a sampled interval $I$ of range $\{[a, b], [a, \infty], [-\infty, a]\}$, where $a \sim \mathcal{U}_{[-100,100]}$. The length of a finite interval is sampled from $\mathcal{U}_{[5,20]}$. Corruption is defined as noise $X \in \mathcal{N}(\mu, 0.01)$ replacing the function values on $I$. We choose $\mu$ as the mean value of $f(x)$ on its domain. The interpreter is prompted to discover the corrupted interval, if any, and to return $a$ and $b$. $15\%$ of the functions $f(x) \in \mathcal{A}$ in the benchmark dataset are corrupted on part of their domain.

**Approximation** of atomic functions is implemented using a two-layer neural network (MLP) with a ReLU non-linearity. For $15\%$ of the functions in the dataset, we train an MLP for 10k epochs on 10k points uniformly sampled on its domain bounded by $(-100, 100)$. Trained MLPs are provided in the benchmark dataset and loaded by the corresponding function during interpretation [2].

## 2.2 Functions on strings

We build on a long history of using toy problems on strings [Hofstadter et al., 1995, Hofstadter, 1995, Mitchell, 2021] and simple visual matrices [Lovett and Forbus, 2017, Wang and Su, 2015, Carpenter et al., 1990, Chollet, 2019] to test the ability of a system to reverse-engineer underlying symbolic operations. As this release of **FIND** is designed to evaluate language-based interpretability systems, we focus on functions with text inputs and procedurally generate a set of functions on strings with different levels of complexity. The benchmark dataset contains 1000 string functions sampled from the **FIND** API, representing both atomic functions (30%) and compositions (70%).

**Atomic functions** include common string manipulation operations such as concatenate, replace, and reverse, and implementations of copycat problems from Hofstadter et al. [1995], such as shift_last ($abc \mapsto abd$). Example atomic string operations are shown in Table 1, and the full set can be found in the **FIND** API.

**Composition** of atomic functions $f(x)$ and $g(x)$ is defined as $(g \circ f)(x) = g(f(x))$. The **FIND** API supports compositions where $(g \circ f)(x)$ is well-defined (i.e. function pairs where $(g \circ f)(x) = x$ are excluded). Example composition functions are shown in Table 1.

## 2.3 Synthetic neural modules

**FIND** includes a set of synthetic neural modules that perform word-level tasks built from Wikidata properties [Vrandečić and Krötzsch, 2014]. We construct two types of text modules: functions involving lexical semantics and functions involving factual relations. **FIND** implements synthetic neural modules on text inputs using a single pretrained language model (Vicuna-13B) instructed to apply different rules to inputs in response to queries (See Figure 2). Vicuna [Chiang et al., 2023] is a fine-tuned LLaMA model licensed under an Apache License 2.0 and open-sourced for non-

---

[2]We provide an LM baseline where MLPs are treated as black boxes during interpretation, similar to the other functions in the dataset. An alternate paradigm could allow probing of internal activations and parameters.

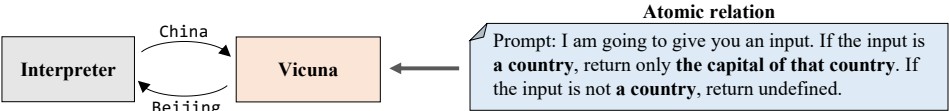

**Figure 2: Synthetic neural module implementation.** Vicuna acts as a backbone that generates the outputs of neural modules. Each function provides instructions to Vicuna that mimic the behavior of either a single neuron ("return an association score with a predefined concept") or a more complex module ("map inputs to outputs according to a a predefined mapping"). The interpreter then interacts with Vicuna as a black-box function.

commercial use. Each function in the dataset is built to interact with Vicuna to return the relevant output to the `interpreter`, which tests the function on different inputs to recover the underlying rule. We find that Vicuna reliably implements functions expressed in prompts (reliability scores are provided for each function type; see Appendix for full evaluation).

### Type 1: Entities

**Atomic entity functions** $f_e(x)$ compare input text to reference concepts (*entities*) drawn from metaclasses of Wikidata properties related to a particular concept or subject area (e.g. *related to food and eating*) and return a value indicating how associated an input is with the entity. Table 1 shows example entities, and the complete list of 140 atomic entities included in **FIND** is provided in the Appendix. The task of an `interpreter` evaluating an entity function is to recover and describe the underlying concept, or, what all of the inputs that produced high output values had in common. For example, if the entity is *plants and botany*, $f_e(\texttt{garden})$ and $f_e(\texttt{tree})$ should return high values, while $f_e(\texttt{car})$ should return a low value. This task is relevant to lines of work that automatically summarize maximally activating exemplar sets to characterize neuron function [Bau et al., 2020, Mu and Andreas, 2020, Hernandez et al., 2022, Bills et al., 2023, Singh et al., 2023], generalized to a setting where the `interpreter` must also produce the data.

To implement each function, we instruct Vicuna to synthesize a binary response to `interpreter` queries, corresponding to whether an input word is associated (`return 1`) or unassociated (`return 0`) with the reference entity (see the Appendix for Vicuna prompts). The function output $s = f_e(x)$ is then a continuous scalar value representing Vicuna's internal probability of 1 being the response token, between a choice of `0` and `1`. Specifically, $s = p_1/(p_0 + p_1)$, where $p_i = e^{\text{logit}_i}$ and $\text{logit}_i$ represents Vicuna's output logit for token $i$. To validate that Vicuna can reliably identify associations between inputs and reference entities, we collect a human-labeled set of concepts $\hat{x}_j$ associated with each entity $e_j$ in the dataset, and compute $\hat{s} = f_{e_j}(\hat{x}_j)$. The mean value of $\hat{s}$ across all 140 entities in **FIND**, for 10 human-annotated concepts per entity, is 0.85. We compute the same score for 10 distractor concepts per entity, sampled from the list of human annotations of other entities. The mean score for the distractors is 0.08. See the Appendix for full experiment details.

**Composed entity functions** mimic the behavior of neurons inside deep networks that are selective for multiple concepts [Fong and Vedaldi, 2018, Olah et al., 2020, Mu and Andreas, 2020]. We construct compositions of entity functions by sampling two entities from the dataset and instructing Vicuna to return 1 if the input value to the function is associated with either entity. 60 composed entity functions are included in the **FIND** benchmark.

### Type 2: Relations

More complex functions inside language models learn factual associations [Meng et al., 2022]. To mimic the behavior of these modules, we construct a set of relation functions that map inputs to outputs according to a real-world factual association (for instance, river $\mapsto$ length or gemstone $\mapsto$ color). Atomic relations are drawn from Wikidata properties included in the Wikidata dump provided by Sorokin and Gurevych [2018]. Table 1 shows example relation functions; a full list is provided in the Appendix. Again we use Vicuna to implement a black box function that applies the rule in the relation to `interpreter` inputs (we verify that Vicuna returns factually correct answers, see Appendix). Figure 2 shows the prompt template for Vicuna and an example relation function.

**Relation functions with bias** include corruption to part of the function domain. **FIND** tests whether an interpretability model can uncover which parts of the function domain have been corrupted. Corrupted relation functions return `undefined` for a small region of the function domain; for example, we corrupt the country $\mapsto$ capital relation on the subdomain `Asia` by prompting Vicuna to return `undefined` for inputs in the subdomain.

# 3 Evaluation protocol

The **FIND** protocol evaluates `interpreters'` natural language descriptions of functions in all categories. In categories where domain corruption is introduced (numeric functions and factual relations), `interpreters` are evalauted on their ability to return a description of the domain of the function, indicating any subdomain where the function behaves differently. In the mathematical reasoning and strings categories, `interpreters` are evaluated on their ability to output code that approximates the function. Section 4 provides details on the interpretation procedures we evaluate and how they are instructed to engage with functions.

## 3.1 Evaluation metrics

To evaluate the accuracy of function descriptions produced by an `interpreter`, we use success indicators for individual function interpretations and calculate average success rates across all functions. For functions where `interpreters` write code approximating the function (numeric and string functions), we score the accuracy of the interpretation by running the `interpreter`'s code on a representative test set and comparing the result to execution of the ground-truth function. To evaluate language descriptions of functions, we judge how well the `interpreter`'s description agrees with ground-truth function behavior (for string functions and synthetic neural modules). Below we describe success indicators for each function category in more detail.

**Evaluating code-based interpretations.** Running interpretation on the **FIND** benchmark produces descriptions of numeric and string functions in code as well as natural language. We measure explanation accuracy by comparing performance of the `interpreter`'s code to the ground-truth function implementation on a test set. For numeric functions, we compute a normalized mean-squared error $\mathbb{E}[(f(x) - g(x))^2]/\mathbb{E}[f(x)^2]$ for $[-128 \leq x \leq 128]$ between the ground truth **FIND** function $f$ and the `interpreter`'s implementation $g$, and regard a successful estimation as one with Normalized MSE (NMSE) $< 0.1$. For string functions, we run an exact-matching binary test ($f(\texttt{<string>}_i)\texttt{==}g(\texttt{<string>}_i)$) on a set of 10 test inputs per function.

**Evaluating language-based interpretations.** We define a "unit testing" protocol where an LM `evaluator` selects which of three input-output (I-O) pairs corresponds to a language description of a function. To judge the `interpreter`'s accuracy, we provide the estimated function description (*e.g.* $f$ : country $\mapsto$ capital) to the `evaluator`, as well as three example I-O pairs: one execution of the ground-truth function (*e.g.* Germany $\mapsto$ Berlin) and two randomly sampled distractors (I-O pairs for other **FIND** functions of the same type; *e.g.* Germany $\mapsto$ Europe, ruby $\mapsto$ red). The evaluator selects which I-O pair matches the functionality of the description from the `interpreter`. If the language description is accurate, the `evaluator` should select the ground-truth I-O pair as the best match. We run this procedure on language descriptions of string functions and synthetic neural modules for a test set of ten different triplets per function. Test sets are constructed to reveal representative behavior of each function using inputs inside and outside of the function domain. For relations corrupted on part of their domain (*e.g.* $f$ : country $\mapsto$ capital, unless the country is in South America), the test set includes two ground truth examples from the corrupted subdomain (*e.g.* Peru $\mapsto$ undefined, Argentina $\mapsto$ undefined). For entity functions that compute similarity to a reference concept (*e.g.* Greek Mythology), we provide the `evaluator` with only input concepts instead of I-O pairs (*e.g.* Athena, skiing, GPU), and ask which concept the function described by the `interpreter` is selective for. As test cases can be designed to isolate specific function behaviors, we find the unit testing protocol to be more sensitive to small differences in function descriptions (*e.g.* "transportation" vs "road transportation") than other description-matching methods, such as having an LM directly grade the agreement between descriptions (see the Appendix for more details).

We finetune Vicuna-13b (`vicuna-evaluator`) to perform the unit testing task and select representative samples matching descriptions of functions in the **FIND** dataset. LM-judges have been shown to be scalable and accurate surrogates for human judgments, which are otherwise expensive to obtain [Zheng et al., 2023]. While proprietary models such as GPT-4 demonstrate strong agreement with humans, using such models to compare `interpreter` performance on a benchmark task incurs prohibitive costs associated with API access and poses reproducibility challenges as model versions are deprecated. We find that `vicuna-evaluator` matches *ground-truth* function descriptions to representative inputs and outputs more accurately than GPT-4, GPT-3.5, or pretrained Vicuna-13b (see Appendix for evaluation). Furthermore, `vicuna-evaluator` makes judgments that are highly correlated with those of human subjects performing the same task (see Appendix for experiment

details). The fine-tuned `vicuna-evaluator` checkpoint and training dataset can be downloaded from the **FIND** repository. Training details are provided in the Appendix.

**Extensions: Targeted evaluation.** For users of the **FIND** benchmark that produce function interpretations in a structured format, this benchmark enables other evaluations targeted at specific end-use cases for interpretability tools. For example, researchers may use **FIND** to explicitly evaluate whether `interpreters` can pick out the portion of the domain that is corrupted, whether they can identify components of composed functions, or whether they identified the noise model.

## 4 Automated Interpretability Methods

**FIND** can be used to evaluate any `interpreter` system that has the ability to execute Python scripts autonomously at the command line. As a first demonstration, we evaluate several approaches inspired by recent work [Hernandez et al., 2022, Bills et al., 2023, Singh et al., 2023] that use pre-trained language models to perform interpretation. We run experiments using the OpenAI Python API. A custom plugin equips the `interpreter` model with the command PYTHON(function.py input) that it can use to call functions on its own selection of inputs. Scripts for reproducing these baselines and complete interpretation dialogues are available in the **FIND** code repository.

We evaluate three different interpretation methods. (i) *Non-interactive*: interpretability tasks often involve descriptions of precomputed exemplars of function behavior. In this setting, the `interpreter` is equipped with a fixed set of inputs to use for probing the functions, and is prompted to produce a description based only on function outputs for these exemplars, mirroring the non-interactive description paradigm used by MILAN [Hernandez et al., 2022] and recent MILAN-type LM analysis approaches [Bills et al., 2023]. (ii) *Automated Interpretability Agents*: LM-based AIAs are prompted to interact with the functions in **FIND**. The `interpreter` serves as an agent and runs PYTHON(function.py input) on inputs it selects. In the base setting, the agents' search process is not initialized with pre-selected inputs; rather, the AIA forms hypotheses by itself, generates data to test its current hypothesis, and interactively updates it based on observations of function outputs. Figure 3 shows complete interactions between a GPT-4-based AIA and example functions in two categories. At the beginning of the interaction, the agent is prompted to run experiments on the functions to produce data that it can use to explain function behavior. Prompts for two categories of functions are reproduced below. (iii) We also evaluate a combination of both approaches (AIA + MILAN). In this setting, the interpretability agent is prompted to begin interpretation with a provided set of exemplars, and then interactively continues the interpretation procedure.

All interpretation methods use off-the-shelf LMs (GPT-4 [OpenAI, 2023], GPT-3.5 (`GPT-3.5-turbo`) [Brown et al., 2020], and Llama-2 (`Llama-2-13b-chat`, `Llama-2-70b-chat`) [Touvron et al., 2023]) as `interpreters`, and operate without a human in the loop. AIA-based interactions show structured interpretation procedures, even when the final explanation is not fully accurate (see the predicted numeric function in Figure 3). The AIA makes and tests informed hypotheses about the functions: for example, testing densely spaced points in a specific interval to reveal a "separating point" in the behavior of the numeric function, and testing different forms of transportation to determine whether a synthetic neuron is selective for land, sea, or air travel (Figure 3). Once the AIA has gathered sufficient data and produces a final description, the run stops automatically. The Appendix further describes experimental settings.

**Prompt for numeric functions:**

```
Hi, your job is to interpret a function f(x) that is implemented in the {function.py} script. f(x) takes
numerical inputs. All you can do is call f(x) on a value of x by running PYTHON({function.py} value). Try
to describe the function implemented by f(x) by running experiments on it. You can call
the function on multiple inputs at a time by running PYTHON({function.py} value1 value2 value3...).
You can run the function multiple times on the same input. We encourage testing a large range of values
before writing a description. You can test values between [-128 and 128]. The output could be
noisy and your goal is to find a good simple description of f(x) that explains most of the function
behavior. f(x) may combine multiple different operations. There may be an interval inside the domain
where the function behaves differently. If that is the case, describe that interval. Your description of
the function should have three parts. First,
[DESCRIPTION]: Describe what the function is doing in language.
[DOMAIN]: What is the domain of the function? Is there an interval where the function displays unexpected
behavior? What are the numeric bounds of that interval? What is going on there?
[CODE]: Write Python code that approximates the behavior of the function.
```

**Prompt for synthetic neural modules (entities):**

> Hi, your job is to interpret a function f(x) that is implemented in the {function.py} script. f(x) takes
> word inputs and returns a value between 0 and 1, where a value closer to 1 indicates the association with
> a particular concept. Your task is to figure out what that concept is. f(x) may be selective for more than
> one concept. If that is the case, describe all concepts that f(x) is selective for. Try to describe the
> function implemented by f(x) by running experiments on it. We encourage you to test a diverse and random
> selection of words: spanning many different categories, including both common and obscure words, concrete
> and abstract words. You can call the function on multiple inputs at a time by running PYTHON({function.py}
> word1 word2 word3 ...). We encourage running many tests before writing a description. The goal is to find
> a good simple description of f(x) that explains most of the function behavior. Your description of the
> function should be in the following form. [DESCRIPTION]: Describe what the function is doing in language.

**Numeric function**

$$f(x) = -4.5 * \tanh((2\pi/5.9)(x - 5.7)) + 5.2$$

**AIA (GPT-4):** To start the investigation, let's test the function at different points following these steps: 1. A value less than -128 (outside the allowed domain), 2. A value of -128, 3. A value between -128 and 128 (e.g. 0) 4. A value of 128, 5. A value greater than 128 (outside the allowed domain) PYTHON(FUNCTION -129 -128 0 128 129)

>> RESPONSE:(-129, 9.7) (-128, 9.7) (0, 9.69995) (128, 0.7) (129, 0.7)

**AIA (GPT-4):** The output with values outside the allowed domain suggests that the function may return a fixed value (9.7) below -128 and another fixed value (0.7) above 128. Let's test the function at different points within the domain to gather more data: PYTHON(FUNCTION -110 -100 -50 50 100 110)

>> RESPONSE:(-110, 9.7) (-100, 9.7) (-50, 9.7) (50, 0.7) (100, 0.7) (110, 0.7)

**AIA (GPT-4):** Based on the input-output pairs, we can assume that the function behaves differently for values below and above some threshold between -50 and 50. Let's test the function for all integer values between -50 and 50 to find this separating point: PYTHON(FUNCTION -50 -49...5 6 7 8 ...49 50)

>> RESPONSE:(-50, 9.7) (-49 9.7) … (5 8.05) (6 3.81) (7 1.23) (8 0.76) … (49 0.7) (50 0.7)

**AIA (GPT-4):** Based on the results for integer values between -50 and 50, we can observe the following: - For x ≤ -6, f(x) = 9.7 - For -6 < x ≤ 0, f(x) decreases non-linearly, and there is some noise. - For 0 < x ≤ 50, f(x) = 0.7 To describe the function:
[DESCRIPTION]: f(x) returns a value of 9.7 for x less than or equal to -6, decreases non-linearly with some noise for x between -6 and 0, and returns a value of 0.7 for x greater than 0.
[DOMAIN]: The unexpected behavior interval is when -6 < x ≤ 0. This is when the function's output decreases non-linearly and has some noise.
[CODE]:
```python
def approximate_function(x):
    if x <= -6: return 9.7
    elif -6 < x <= 0:
        return 9.7-(9.7-0.7)*((x + 6)/6)**2
        # Assuming a simple polynomial
        decrease, this can be further improved
    else: return 0.7
```

Please note that the approximation function may not cover every intricate detail of the real function, and can be further refined for the interval between -6 and 0.

**Synthetic neural module**

ENTITY: road transport

**AIA (GPT-4):** Let's start by testing a few random words with the FUNCTION command. PYTHON(FUNCTION cat dog fish tree car house)

>> RESPONSE:(cat, 0.0006) (dog, 0.0013) (fish, 0.0003) (tree, 0.0013) (car, 0.9777) (house, 0.0007)

**AIA (GPT-4):** These outputs seem to have a wide range of values. Let's try some more words to find a pattern. PYTHON(FUNCTION love hate happiness sadness bicycle airplane)

>> RESPONSE:(love, 0.0005) (hate, 0.0005) (happiness, 0.0003) (sadness, 0.0004) (bicycle, 0.4149) (airplane, 0.0014)

**AIA (GPT-4):** Interesting. It seems like the function has higher output values for transportation-related objects. Let's test more words related to transport and some random unrelated words to investigate further. PYTHON(FUNCTION train bus boat walking running swimming lion television)

>> RESPONSE:(train, 0.1294)(bus, 0.9763)(boat, 0.0004) (walking,0.0008)(running, 0.0018)(swimming, 0.0002) (lion, 0.0004)(television, 0.0002)

**AIA (GPT-4):** Based on the given test results, it seems that the function is selective for objects related to ground transportation, specifically vehicles. Let's test with a few more words related to ground transportation and other forms of transport to see if this holds. PYTHON(FUNCTION motorcycle scooter helicopter submarine spaceship)

>> RESPONSE:(motorcycle, 0.9519) (scooter, 0.5351) (helicopter, 0.0011) (submarine, 0.0012) (spaceship, 0.0005)

**AIA (GPT-4):** [DESCRIPTION]: The function f(x) is selective for **ground transportation vehicles**. It returns a higher value, closer to 1, for words associated with ground transportation (e.g., car, bus, motorcycle, scooter) and lower values, closer to 0, for unrelated words or other forms of transportation (e.g., helicopter, submarine, spaceship).

**Figure 3: AIA function interpretation with GPT-4.** The interpretability agent is able to conduct experiments that reveal the behavior of an unseen function, by making hypotheses, selecting inputs that produce informative data, and updating hypotheses in light of new information. We include the full dialogues for two example functions: a numeric function involving a hyperbolic tangent (left), and a synthetic neural module that returns a continuous value between zero and one indicating input association with the ground truth concept *road transport* (right). Numeric function and synthetic neural module prompts were used at the beginning of the respective conversations, and have been omitted from the figure for simplicity. We overlay a plot of the predicted numeric function and the real function for comparison. Points on the real function that GPT-4 sampled are marked. Final explanations are compared to ground truth functions for evaluation (See Table 2).

**Table 2: Interpretation success rates.** For each function type we report the successful estimation rate **(higher better)** based on different indicators, and with different experimental settings (*e.g.* initialization with exemplars).

| | Code (exact match) | | Language (unit test) | | | | | | |
| | Numeric | Strings | Strings | Entities | | | Relations | | |
| | AIA | AIA | AIA | AIA | MILAN | AIA +MILAN | AIA | MILAN | AIA +MILAN |
|---|---|---|---|---|---|---|---|---|---|
| Llama-2-13b-chat | 0 | 0 | 0.33 | 0.34 | 0.54 | 0.58 | 0.46 | 0.45 | 0.42 |
| Llama-2-70b-chat | 0.01 | 0.01 | 0.33 | 0.34 | 0.61 | 0.62 | 0.47 | 0.44 | 0.46 |
| GPT-3.5 | 0.12 | 0.13 | 0.66 | 0.39 | 0.81 | 0.88 | 0.37 | 0.64 | 0.68 |
| GPT-4 | **0.33** | **0.23** | **0.82** | **0.56** | **0.89** | **0.89** | **0.78** | **0.74** | **0.92** |

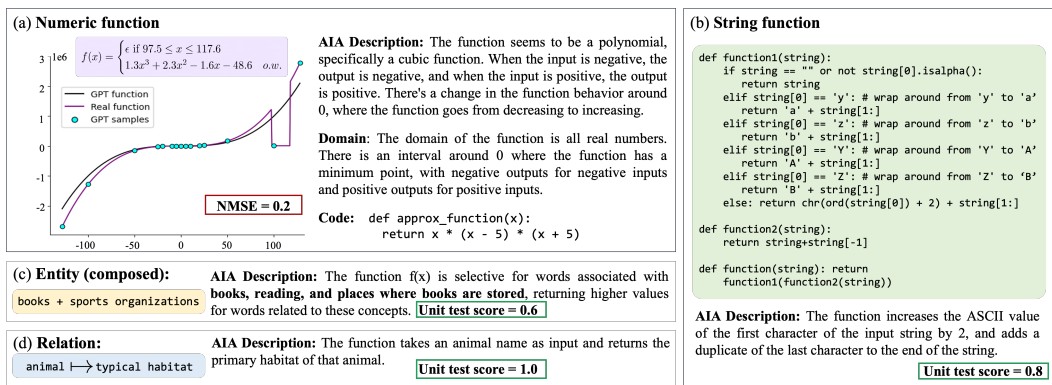

**Figure 4: AIA (GPT-4) interpretations.** Examples from all **FIND** categories, with evaluation scores (NMSE and unit test) marked in red if below the success threshold (NMSE> 0.1, unit test< 0.33) and green otherwise.

## 5 Results

We evaluate the AIA method as well as the non-interactive baselines with different off-the-shelf LMs. Results are summarized in Table 2 and example function descriptions are shown in Figure 4. Additional example interpretations of functions in all categories are provided in the Appendix.

**GPT-4 is a stronger interpretability agent than GPT-3.5 and Llama-2.** Success rates for all function categories are reported in Table 2 (AIA columns). The GPT-4 interpretability agent achieves universally higher success rates than GPT-3.5 and Llama-2 (which often score at chance value). In fact, we find that Llama-2 often invents the output of the function without executing it (see examples in Appendix). This is also reflected in the mean length of interpretation dialogues, which are 1.4, 1.4, 2.1, and 4.1 for Llama-2-13b-chat, Llama-2-70b-chat, GPT-3.5 and GPT-4 respectively (we count the number of `interpreter`-function interactions). We additionally observe that interpretations of string functions receive significantly higher scores using the unit testing protocol compared to string-matching outputs of estimated code. As unit testing selects for representative examples of function behavior and not exact string-matches, the procedure is more forgiving of less specific descriptions, or descriptions with minor inaccuracies (see Appendix for additional discussion of unit testing limitations).

**Interactive vs. non-interactive.** In addition to AIAs, we evaluate two other interpretation methods: MILAN [Hernandez et al., 2022] where the `interpreter` produces a description based only on function outputs for a given set of exemplars, and a combination of AIA + MILAN, where the same set of exemplars is used to initialize the interpretation session, and the agent is subsequently permitted to perform additional experimentation. For both settings, we sample the exemplars (two related to the function, eight distractors) from a human-constructed list of inputs associated with each function (see Appendix for details). In the non-interactive MILAN setting, we observe a small improvement in performance over uninitialized AIAs; GPT-4 interpretation performance improves from 0.56 to 0.89 on entity functions, and decreases slightly on relations, from .78 to .74. Initializing AIAs with MILAN exemplars and allowing additional experimentation dramatically boosts the performance of GPT-4 and GPT-3.5 agents (and also slightly improves Llama-2 performance), as shown in Table 2 and Figure 5 (see *init.*). Notably, when initialized with exemplars, GPT-3.5 exhibits performance interpreting entity functions comparable to GPT-4. These results suggest that off-the-

shelf LM agents are limited by breadth of search. Indeed, LMs tend to start sampling with simple words (*e.g.* `apple, dog, car`) which do not reveal function behavior for highly specific reference entities (*e.g. The New York Times, arachnids and arachnology*). We view exemplar computation as one of many "tools" that an interpretability agent could use, and hope that this benchmark will drive exploration of additional tools (*e.g.* example synthesis) as part of automated interpretation methods. Procedures that combine initialization and interactive experimentation could improve the efficiency of existing labeling approaches that use large fixed datasets [Bau et al., 2017] to precompute maximally activating inputs, and potentially also surface novel behaviors not captured in predefined sets of exemplars.

**Which functions can LMs recover?** Figure 4 shows examples of AIA interpretations that successfully explain the behavior of some functions (*e.g.* cases (b),(d)), but fail to fully characterize more complex functions (*e.g.* cases (a),(c)), also reflected in some per-subcategory success scores (Figure 5). This is a limitation of using off-the-shelf LMs as agents: they may miss small corruptions to part of the domain (Figure 4a), which in real-world interpretability settings could stem from bias in the training set. LMs could be outfitted specifically for interpretability with additional tools (*e.g.* for sampling) and further improved by fine-tuning.

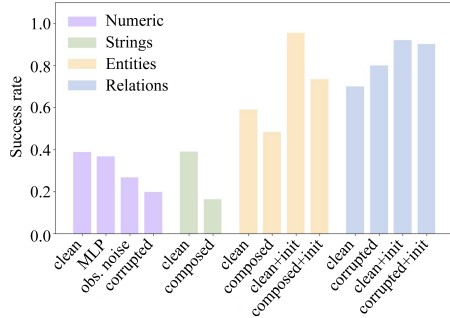

**Figure 5: AIA interpretation scores by subcategory (with GPT-4).** Complex functions are usually more difficult to interpret than atomic functions. String functions are evaluated using the string-matching indicator.

# 6 Related work

**Explanation evaluation methods** have previously benchmarked salience methods according to their ability to recover ground-truth segmentations [Zhang et al., 2018, Selvaraju et al., 2017, Fong and Vedaldi, 2017, Yang and Kim, 2019], or identify inputs that causally affect outputs [Zeiler and Fergus, 2014, Petsiuk et al., 2018, Wagner et al., 2019, DeYoung et al., 2020]. Explanation benchmarks have also evaluated correlation with system performance [Adebayo et al., 2018, Casper et al., 2023] or human understanding of decisions [Kim et al., 2022]. Our benchmark differs because we evaluate global explanations of black box functions instead of evaluating local explanations of decisions.

**Model interpretability metrics** have quantified the degree of model interpretability, *i.e.* by measuring how closely deep network features match a human-labeled concept [Bau et al., 2017, Kim et al., 2018, Goh et al., 2021, Wu et al., 2021, Burgess et al., 2018, Mu and Andreas, 2020, Geva et al., 2021]. While these methods can measure disentanglement in model representations, they are only as good as their ability to identify interpretable features. We tackle this problem by providing a benchmark for interpretation methods themselves, rather than the models that they explain.

**Full-text explanation systems** provide natural-language explanations for black box systems or their features [Hendricks et al., 2016, Camburu et al., 2018, Ehsan et al., 2018, Kumar and Talukdar, 2020, Hernandez et al., 2022]. Recent efforts exploit the capabilities of LMs to explain data directly, but these works utilize only tiny evaluation benchmarks, including 19 synthetic neuron puzzles in Bills et al. [2023] and 54 ground-truth module topics in Singh et al. [2023]. Motivated by the promise of this LM-driven approach and the need to quantify performance, our work provides a comprehensive benchmark of full-text black-box explanation systems.

# 7 Conclusion

We introduce **FIND**, a new benchmark for evaluating automated interpretability methods. Baseline results show early evidence that agents built from advanced LMs can construct hypotheses and experiments to validate them, supporting the suitability of LMs as general-purpose interpretability backbones. However, we find many functions that LM agents cannot sufficiently explain, suggesting augmentation with additional tools will be necessary for robust automation of interpretability tasks.

## Acknowledgments and Disclosure of Funding

We are grateful for the support of the MIT-IBM Watson AI Lab, the Open Philanthropy foundation, an Amazon Research Award, Hyundai NGV, ARL grant W911NF-18-2-0218, the National Science Foundation under Grant Nos. 2212310 and 2238240, the Zuckerman STEM Leadership Program, and the Viterbi Fellowship. The funders had no role in dataset design, experimental design or analysis, decision to publish, or preparation of the manuscript. The authors have no competing interests to report.

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

# Appendix

Supplemental Materials for **FIND: A Function Description Benchmark for Evaluating Interpretability Methods**. This appendix provides hosting, access, and maintenance details, and additional information about the FIND dataset and experiments described in the main paper.

## A    Limitations

We view FIND as a test of necessary, but not sufficient, capabilities for automated interpretation. The ultimate test of these interpretation methods' effectiveness must be their ability to generate actionable insights about real models, which FIND does not evaluate. But clean, simple benchmarks with ground-truth answers have been a major driver of more general capabilities in LMs, and we hope that FIND can play a similar role in interpretability research. Additionally, the current release of FIND only includes black-box interpretation problems. This style of problem is relevant to many existing automated procedures such as NetDissect [Bau et al., 2017] and MILAN [Hernandez et al., 2022], which treat single neurons as black boxes; however, most interpretability work occurs in a white-box setting. We intend for FIND to be extended in the future to incorporate white-box function interpretation problems, including descriptions of individual components of neural circuits (the IOI circuit from Wang et al. [2022] is just one example), some of which could be represented as the types of composition problems that are already included in FIND, but where `interpreters` access and label individual sub-computations inside composed functions.

## B    Ethics statement

FIND includes entities drawn from Wikidata that represent real-world concepts like *video games*, *paleontology*, and *airports*, as well as more potentially sensitive topics like *The Holocaust*, *World War II*, and *disasters* (see Appendix D). We include these topics in FIND because they are relevant to behaviors inside neural networks trained on real-world data, that we want automated interpretation procedures to be able to describe. We additionally note that `interpreters` which test the semantic similarity between input concepts and reference entities may surface surprising or controversial similarity scores that stem from learned biases inside the LM backbone of the synthetic neural modules (*e.g.* men having a higher similarity score with *mathematics* than women). Advanced interpretation procedures would be able to name and describe these biases, but the FIND evaluation protocol currently does not test ability to discover biases other than those included explicitly in FIND (via corruptions to function subdomains).

## C    Additional interpretation examples

Below we provide additional examples of AIA interpretations performed with a GPT-4 agent for each of the FIND categories. For numeric functions, we report the NMSE score (lower is better) marked in green if the score is below the success threshold, or red otherwise. For strings and synthetic neural modules, we report the unit test score (higher is better) marked in green if the score is above chance, or red otherwise. Complete interpretation dialogues are available in the FIND code repository.

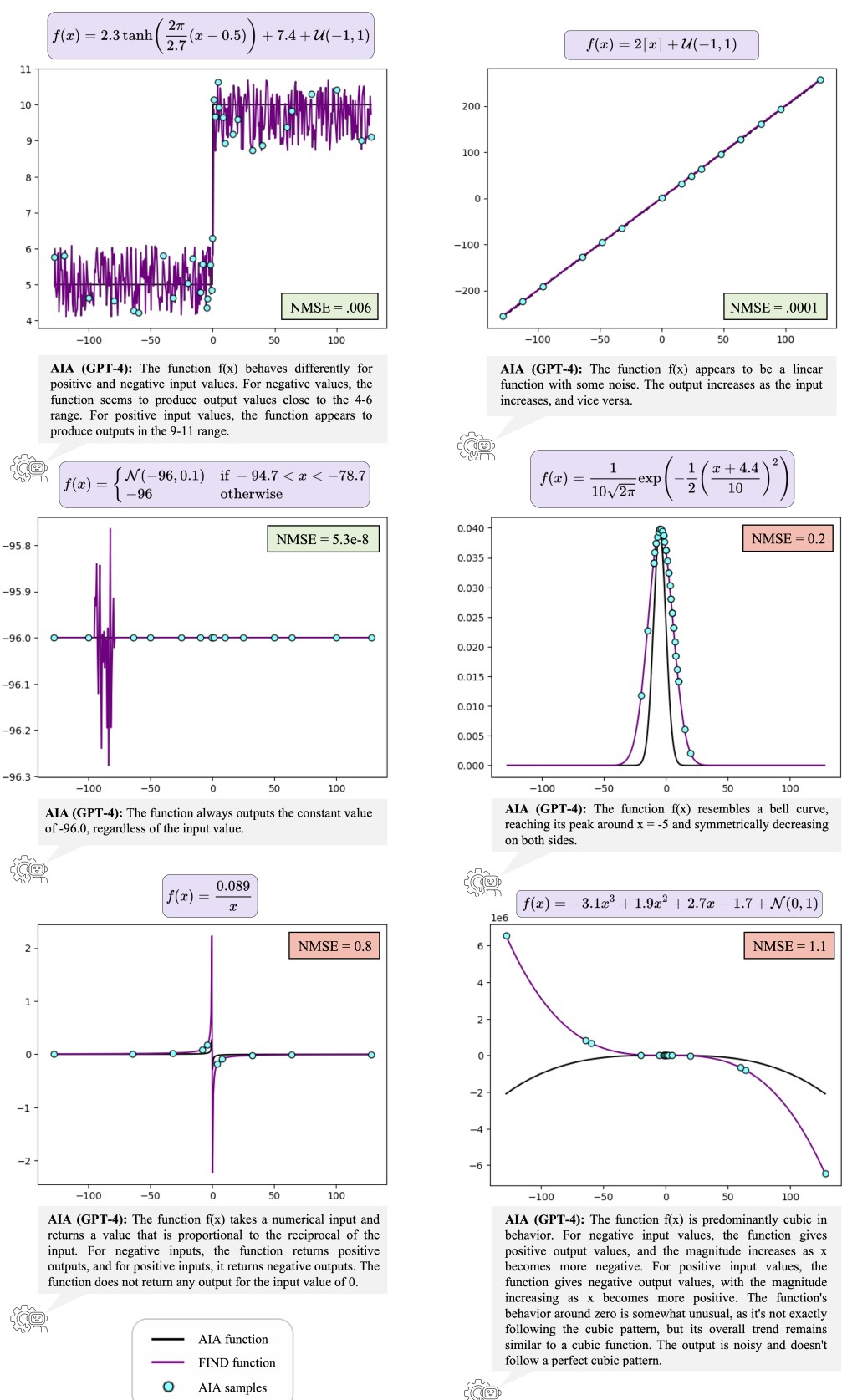

**Figure A1: Numeric function interpretation examples.** Plots compare the code interpretation to the ground-truth function implementation. Points on the **FIND** function sampled by the GPT-4 AIA are indicated. NMSE scores are marked in green if the score is below the success threshold (0.1), or red otherwise.

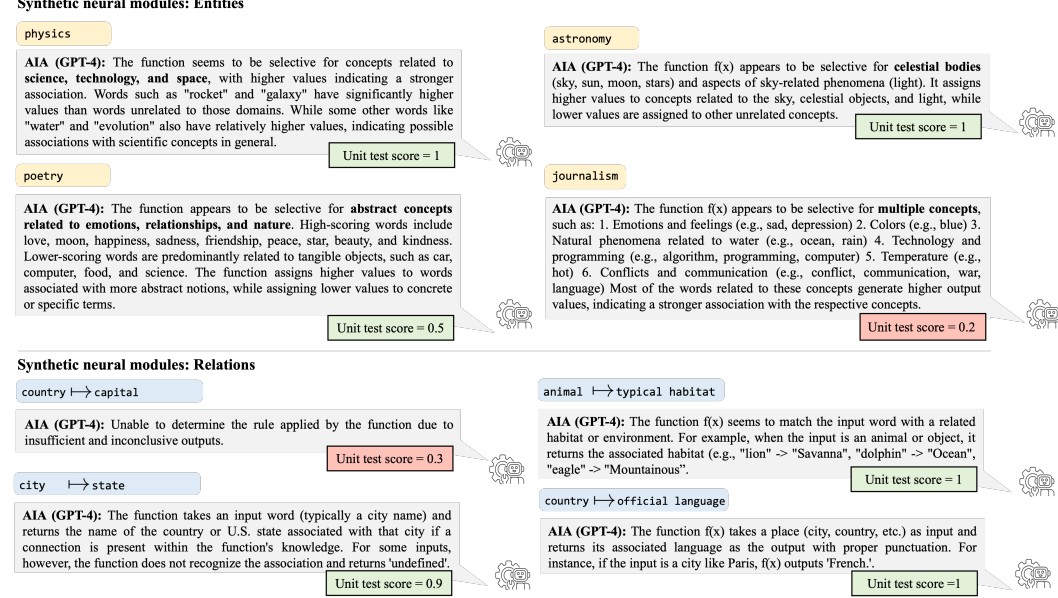

**Figure A2: Synthetic neural module interpretation examples.** Unit test scores are marked in green if the score is above chance (0.33), or red otherwise.

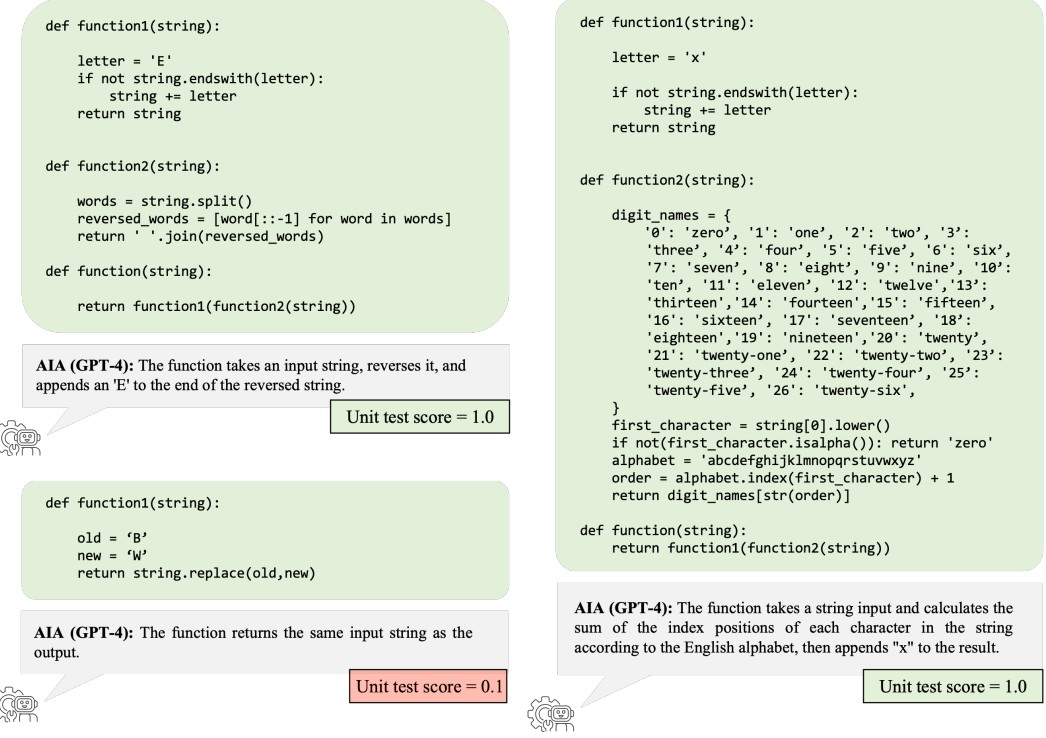

**Figure A3: String function interpretation examples.** Unit test scores are marked in green if the score is above chance (0.33), or red otherwise.

# D   FIND documentation

The FIND API, benchmark dataset, and associated metadata are open-sourced under an MIT license and available at: https://github.com/multimodal-interpretability/FIND.

The FIND repository contains the utilities necessary for reproducing benchmark results for the LM baselines reported in the paper, and running and evaluating interpretation of the FIND functions with other `interpreters` defined by the user.

The FIND dataset itself is hosted on Zenodo with DOI: 10.5281/zenodo.8034162. The GitHub README provides instructions for downloading the dataset file into the appropriate directory for use with the FIND utilities.

**Responsibility statement.** The authors affirm that they have full authority to license the dataset under the terms of the MIT License and they accept all responsibilities for the dataset.

**Dataset.** Each function is expressed as an independent `.py` file that can be called by the `interpreter` with the FIND utilities. The benchmark dataset includes:

1. 1000 numeric functions sampled from the FIND API (see section E)
2. 1000 string functions sampled from the FIND API (see section F)
3. 275 synthetic neural modules with an LM backbone (see section G )

An `interpreter` can be tested on a single function or the entire FIND dataset using `collect_interpretations.py`. Most of the functions are self-contained, see one example here (`find_dataset/numeric/f00470/function_code.py`):

```python
import numpy as np
import sys

def function(x):
    return (-4.5*x**4 - 0.7*x**3 - 2.6*x**2 + 0.4*x**1 + 3.9) * \
            (-25.3 * np.where(x > 0, x, x * 0.3) + 5.2)

if __name__ == '__main__':
    outputs = ''
    for arg in sys.argv[1:]:
        x = float(arg)
        try:
            out = function(x)
        except:
            out = 'None'
        outputs += f'({arg}, {out})'
    print(f'Function input - output pairs: {outputs}')
```

For numeric functions that load trained MLPs, the dataset includes a `.pt` file with model weights in the same folder as `function_code.py`. Other functions (synthetic neural modules) that are implemented with an LM backbone rely on calls to a self-hosted Vicuna inference server. For example, this function (`find_dataset/neurons_entities/f00013/function_code.py`) simulates a neuron selective for `works of fiction` by measuring Vicuna output logits:

```python
import openai
import sys
import numpy as np
import torch

openai.api_key = "EMPTY"
openai.api_base = {API_BASE}
model = "vicuna-13b-v1.1"

from transformers import LlamaTokenizer
tokenizer = LlamaTokenizer.from_pretrained({LLAMA_PATH})
zero_token_id = tokenizer._convert_token_to_id('0')
one_token_id = tokenizer._convert_token_to_id('1')

concept = 'works of fiction'

def function(input, sys_prompt):
    messages=[
        {"role": "system", "content": sys_prompt},
        {"role": "user", "content": # Vicuna prompt for function
        f"I am going to give you an input. Return only \"1\" if that input is related to
        {concept}." + "Return only \"0\" otherwise.\n
        Do not return any additional information other than the 0 or 1 answer.\n" +
        f"Input: {input}"}          # input from interpreter
    ]

    completion = openai.ChatCompletion.create(
        model=model,
        messages=messages,
        temperature=0.0
    )

    logprobs = np.array(completion.choices[0].logprobs)
    output_tokens = tokenizer.encode(completion.choices[0].message.content, return_tensors='pt')
    indices  = torch.logical_or(output_tokens == zero_token_id, output_tokens == one_token_id)
    indices = indices.int()
    index_of_choice = indices.nonzero()
    index = index_of_choice[0][1].item()
    index = index-1
    prob_1 = np.exp(logprobs[index, one_token_id]) / (np.exp(logprobs[index, zero_token_id]) +
            np.exp(logprobs[index, one_token_id]))

    return prob_1

if __name__ == '__main__':
    outputs = ''
    prompt = "You are an AI trained to identify words related to input concepts."
    for arg in sys.argv[1:]:
        outputs += f'({arg}, {function(arg, prompt):.4f}) '
    print(f'Function input - output pairs: {outputs}')
```

Calling these functions requires running inference on Vicuna. Instructions for setting up and running a Vicuna inference server can be found in the **FIND** GitHub README. Vicuna is open-sourced (Apache License 2.0) for non-commercial use. Vicuna is also subject to the LLaMA model license (GNU General Public License v3.0), the OpenAI Terms of Use, and the ShareGPT Privacy Practices.

**Reproducibility.** Many of the functions in this dataset are self-contained; the ones that are not can be reproduced with Vicuna-13B-v1.1, which is open-sourced and accessible to research GPUs, or can be run in the cloud (to run the interpretation baselines in this paper, we used 2x NVIDIA GeForce RTX 3090 GPUs). Running the evaluation requires hosting vicuna-evaluator. We provide the finetuned vicuna-evaluator checkpoint for download via the **FIND** GitHub. All scripts for reproducing evaluations reported in the paper are available in the GitHub repository, and we discuss comparisons to other LMs as evaluators in section F.

# E Numeric function API details

The API for generating numeric functions can be found at:
https://github.com/multimodal-interpretability/FIND/src/make_functions/make_numeric.

The library of numeric functions is located in the script math_functions.py, which defines individual functions in the set $\mathcal{A}$ of atomic functions and randomly samples their parameters. We also randomly sample integer scale and bias parameters between $[-30, 30]$ for each function.

15% of the numeric functions in the **FIND** dataset are compositions of two atomic functions $f(x)$ and $g(x)$. To limit the complexity of the compositions, we choose $f(x)$ and $g(x)$ from a subset $\mathcal{A_C}$ of atomic functions, where $\mathcal{A_C}$ = linear, polynomial, step, RELU, constant, ceiling, floor, rectangle, square wave, and combine them using only multiplication and addition.

**FIND** is extensible: the API can be used as a general numeric function generator, allowing the user to make other design choices targeting specific interpretability applications.

## F   String function API details

The API for generating string functions can be found at:
https://github.com/multimodal-interpretability/FIND/src/make_functions/make_strings.

All functions and sampling parameters used to create the **FIND** dataset are articulated in string_functions.py. Like with numeric functions, we define a subset (described in the API) of string functions with lower complexity to use in compositions.

## G   Synthetic neural modules

To validate the reliability of Vicuna as a backbone for the synthetic neural modules, we score its execution of the **FIND** functions on human labels associated with each entity in **FIND**, and ground-truth factual relation pairs (extracted from Wikidata).

### D1   Entities

For each entity, we construct a list of 10 associated concepts (*e.g.* for the *climate* entity, associated words include humidity, temperature, atmosphere). In Table A1 we list all entities included in **FIND**, and report the mean function output across 10 human labels associated with each entity (function output is computed from Vicuna output logits) and the mean function output for 10 labels randomly sampled from the rest of the dataset. A score closer to 1 indicates stronger association between input concept and reference entity. For all entities in **FIND**, Vicuna scores reliably distinguish between associated and unassociated concepts. We additionally show selectivity of each entity function for concepts associated with that entity relative to all other entities in Figure A4.

**Table A1:** Atomic entities and Vicuna scores for Synthetic Neural Modules

| Entity | Same Entity Score | Random Entity Score |
|---|---|---|
| sport organizations | 0.996 | 0.112 |
| corporations | 0.968 | 0.053 |
| elections | 0.670 | 0.012 |
| musical works | 0.978 | 0.110 |
| films | 0.992 | 0.023 |
| climate | 0.810 | 0.009 |
| computer hardware | 0.972 | 0.004 |
| physics | 0.992 | 0.021 |
| religions and beliefs | 0.899 | 0.013 |
| law and justice | 0.995 | 0.104 |
| transport | 0.786 | 0.129 |
| education | 0.783 | 0.122 |
| politics | 0.508 | 0.029 |
| works of fiction | 0.971 | 0.087 |
| games and leisure activities | 0.763 | 0.136 |
| biology | 0.994 | 0.168 |
| mathematics | 0.993 | 0.020 |
| geology | 0.993 | 0.044 |
| government and state | 0.644 | 0.067 |
| food and eating | 0.855 | 0.101 |
| air transport | 0.763 | 0.011 |
| road transport | 0.787 | 0.027 |

| Entity | Same Entity Score | Random Entity Score |
|---|---|---|
| rail transport | 0.672 | 0.001 |
| cycling | 0.846 | 0.011 |
| the New York Times | 0.703 | 0.163 |
| aircraft | 0.737 | 0.003 |
| bodies of water | 0.992 | 0.094 |
| mineralogy | 0.984 | 0.054 |
| occupations | 0.976 | 0.178 |
| professional wrestling | 0.872 | 0.004 |
| quantity indicating a percentage | 0.996 | 0.148 |
| plays | 0.546 | 0.149 |
| horses | 0.964 | 0.005 |
| photography | 0.977 | 0.026 |
| music | 0.965 | 0.038 |
| fashion | 0.898 | 0.154 |
| chess | 0.700 | 0.004 |
| racket sports | 0.828 | 0.129 |
| art | 0.991 | 0.457 |
| disasters | 0.687 | 0.022 |
| spacecraft | 0.915 | 0.018 |
| video games | 0.991 | 0.035 |
| the relationship of an element to its class | 0.893 | 0.708 |
| basketball | 0.767 | 0.009 |
| winter sports | 0.984 | 0.002 |
| American football | 0.849 | 0.036 |
| golf | 0.925 | 0.026 |
| baseball | 0.893 | 0.006 |
| ice hockey | 0.663 | 0.003 |
| women and feminism | 0.819 | 0.120 |
| lighthouses | 0.493 | 0.010 |
| tennis | 0.804 | 0.008 |
| water sports | 0.958 | 0.014 |
| comics | 0.637 | 0.012 |
| algorithms | 0.801 | 0.021 |
| tourism | 0.690 | 0.186 |
| bridges | 0.685 | 0.012 |
| books | 0.774 | 0.111 |
| linguistics | 0.995 | 0.464 |
| utilization and ownership | 0.402 | 0.206 |
| online communities | 0.889 | 0.084 |
| television | 0.851 | 0.023 |
| astronomy | 0.995 | 0.054 |
| disability | 0.637 | 0.023 |
| proteins | 0.399 | 0.072 |
| human anatomy | 0.966 | 0.053 |
| gymnastics | 0.751 | 0.026 |
| architecture | 0.800 | 0.055 |
| sculpture | 0.795 | 0.030 |
| archaeology | 0.836 | 0.083 |
| theater | 0.899 | 0.104 |
| dams | 0.776 | 0.012 |
| birds and ornithology | 0.993 | 0.070 |
| computing | 0.988 | 0.302 |
| time and duration | 0.699 | 0.052 |
| age | 0.832 | 0.040 |
| weapons and military equipment | 0.820 | 0.003 |
| ratios and proportions | 0.446 | 0.155 |
| typefaces and typography | 0.911 | 0.094 |
| burials graves and memorials | 0.878 | 0.010 |
| processes and manufacturing | 0.645 | 0.058 |
| paleontology | 0.934 | 0.054 |
| banking | 0.922 | 0.034 |
| lakes | 0.552 | 0.060 |

| Entity | Same Entity Score | Random Entity Score |
|--------|-------------------|---------------------|
| natural science | 0.970 | 0.359 |
| geography | 0.981 | 0.039 |
| insects and entomology | 0.897 | 0.033 |
| philosophy | 0.991 | 0.168 |
| poetry | 0.991 | 0.071 |
| encyclopedias | 0.642 | 0.350 |
| television shows | 0.914 | 0.095 |
| awards prizes and honours | 0.997 | 0.159 |
| the Middle Ages | 0.871 | 0.053 |
| plants and botany | 0.994 | 0.189 |
| marine biology | 0.996 | 0.059 |
| color | 0.780 | 0.080 |
| airports | 0.712 | 0.003 |
| science | 0.946 | 0.284 |
| anime and manga | 0.969 | 0.094 |
| Christianity | 0.988 | 0.015 |
| Buddhism | 0.993 | 0.002 |
| Greek mythology | 0.949 | 0.013 |
| Judaism and the Jewish people | 0.965 | 0.012 |
| music genres | 0.987 | 0.144 |
| fictional characters | 0.892 | 0.144 |
| rap and hip hop | 0.693 | 0.055 |
| Islam | 0.949 | 0.016 |
| The Walt Disney Company | 0.937 | 0.005 |
| agriculture | 0.982 | 0.104 |
| hiking | 0.384 | 0.018 |
| New York City | 0.969 | 0.113 |
| gardens | 0.896 | 0.093 |
| personality traits | 0.879 | 0.172 |
| camping | 0.728 | 0.065 |
| gender | 0.555 | 0.049 |
| cemeteries and graves | 0.984 | 0.027 |
| London | 0.954 | 0.105 |
| Los Angeles | 0.949 | 0.042 |
| Chicago | 0.959 | 0.047 |
| Paris | 0.789 | 0.017 |
| Berlin | 0.923 | 0.052 |
| sailing | 0.897 | 0.001 |
| swimming | 0.901 | 0.015 |
| the Holocaust | 0.788 | 0.007 |
| arachnids and arachnology | 0.798 | 0.128 |
| musical instruments | 0.997 | 0.042 |
| meteorology | 0.926 | 0.011 |
| disease | 0.787 | 0.010 |
| Ireland | 0.784 | 0.020 |
| libraries | 0.612 | 0.122 |
| museums | 0.848 | 0.058 |
| journalism | 0.883 | 0.020 |
| buildings | 0.795 | 0.153 |
| cryptocurrencies | 0.951 | 0.001 |
| events and news | 0.725 | 0.229 |
| Louisiana | 0.745 | 0.012 |
| revolutions | 0.623 | 0.076 |
| mines and mining | 0.918 | 0.039 |
| Switzerland | 0.947 | 0.050 |
| World War II | 0.986 | 0.064 |
| **Average** | **0.846** | **0.079** |

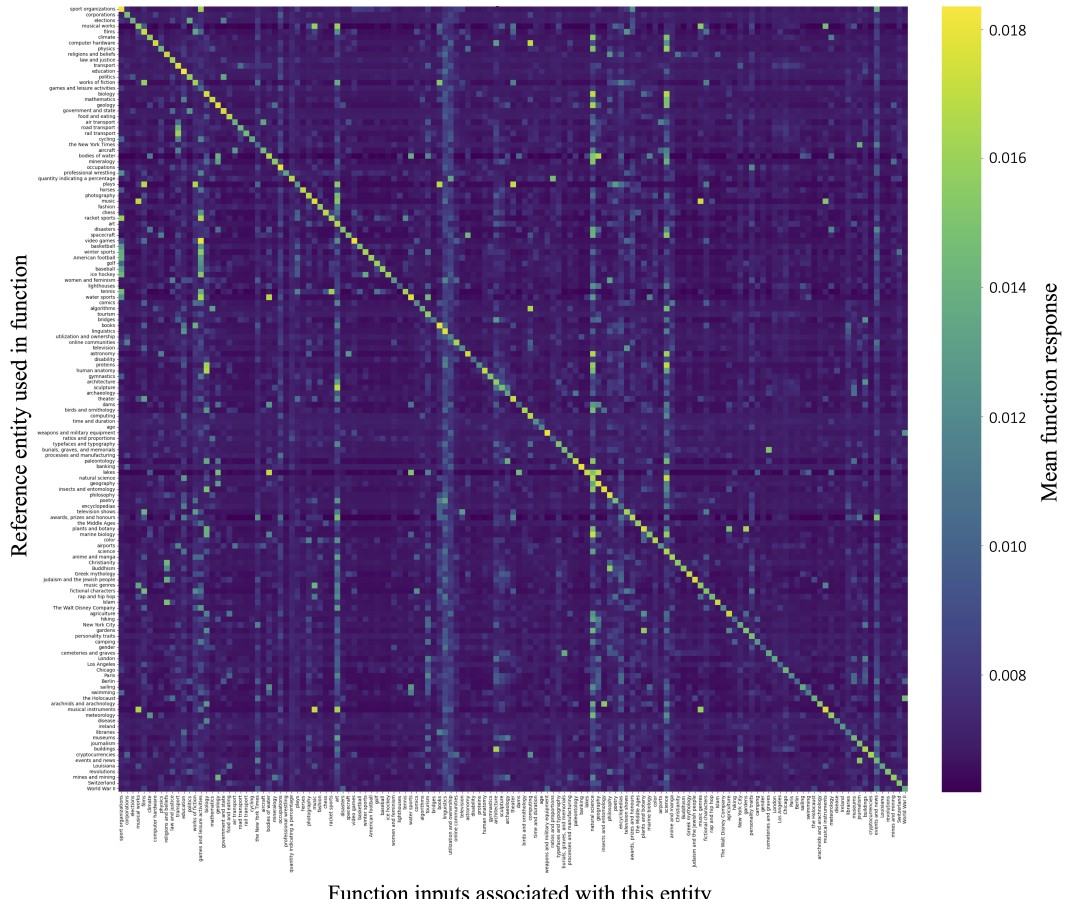

**Figure A4:** Entity functions implemented using Vicuna respond more strongly to concepts related to their reference entity (diagonal) than to concepts related to other entities (off-diagonal). The value in row $i$ and column $j$ corresponds to the average output of the entity $e_i$ function for 10 inputs associated with entity $e_j$. Each row is normalized using softmax. The prompt template used to instruct Vicuna to implement each function is shown in Section G.

**Table A2:** Factual relations and domain corruptions

| Input | Output | Domain corruptions <case 1, case 2> |
|---|---|---|
| country | capital of the country | country is in <Asia, S. America> |
| city | country where the city is located | city is in <Mexico, The U.S.> |
| head of state | country where that person was head of state | head of state is a <man, woman> |
| monument | country where monument is located | monument in <New York City, France> |
| monument | continent where monument is located | monument in <S. America, Australia> |
| city in the U.S. | state where the city is located | city is in <California, Texas> |
| country | official language of the country | country in <N. America, S. America> |
| well-known person | occupation of the person | person is a <woman, man> |
| name of a person | most likely gender of the person | name begins with <T, S> |
| country | continent where the country is located | country is in <Asia, Europe> |
| opera | composer of the opera | opera is in <Italian, German> |
| title of a book | author of the book | book written <pre-1900, post-1900> |
| country | another country on its border | country is in <Africa, N. America> |
| politician | political party of the politician | politician is a <man, woman> |
| river | continent where the river is located | continent is <Africa, Europe> |
| city | time zone of the city | city is in <N. America, Asia> |
| opera | language of the opera | opera is in <English, Italian> |
| animal | main food source of the animal | animal is a <carnivore, herbivore> |
| animal | typical habitat of the animal | animal lives on <land, in the ocean> |
| gemstone | color of the gemstone | gemstone is <red, blue> |
| country | color of the country's flag | country is in <N. America, Asia> |
| river | length of the river | river is in <N. America, Europe> |
| country | maximum elevation in the country | country is in <S. America, Africa> |
| country | height of the tallest skyscraper in that country | country is in <N. America, Asia> |
| country | height of the average person in that country | country is in <Europe, S. America> |

## D2 Relations

**FIND** includes 75 synthetic neural modules that map inputs to outputs using real-world factual relations drawn from Wikidata properties. The set of relation functions is built from 25 atomic relations (*e.g.* country ↦ capital) and two domain corruptions per relation, where the function returns `undefined` for a small region of the domain (*e.g.* country ↦ capital EXCEPT if `country in South America`, then return `undefined`). Table A2 lists atomic relation functions included in **FIND**, and the two domain corruptions applied to each function. We evaluate Vicuna accuracy applying relation functions to input concepts by scoring the accuracy of the mapping applied to 10 inputs per function, where inputs are drawn from concepts in Wikidata with the property applied by each relation function. Mean Vicuna accuracy across relations in **FIND** is $91.2\%$.

# E Interpretation

Utilities for reproducing interpretation experiments and adding other, user-defined `interpreters` are provided in the GitHub repository. We report interpretation baselines using GPT-4, GPT-3.5 (`GPT-3.5-turbo`), and Llama-2 (`Llama-2-13b-chat`, `Llama-2-70b-chat`). Interpretation dialogues never exceeded context length of 8K tokens for GPT-4, 4K tokens for GPT 3.5, and 2048 for Llama-2. Full interpretation dialogues from our baselines can be downloaded from https://data.csail.mit.edu/FIND/FIND-interpretations.zip.

## E1 Llama-2 hallucination

We note that Llama-2 often fails to engage with the functions and instead hallucinates outputs before function output values are returned to the interpretation dialogue. Example hallucinations are shown below for `Llama-2-13b-chat` interpreting a synthetic neural module function. We tested different system prompts to encourage Llama-2 to perform the task. The top-performing Llama-2 prompt is reproduced in Section G.

## E2 Sampling initialization experiments (AIA +MILAN)

Initialization experiments were performed for synthetic neural modules where search strategy (initially guessing very simple words) handicapped `interpreter` performance. As described in Section 5 of the main paper, in this setting the `interpreter` was provided with a list of 10 inputs per function to use for initialization (two associated with the function, eight distractors). These exemplars were sampled from the lists of human-labeled concepts associated with each function as described in Appendix G. The set of suggested initial inputs we used for each function is included in the **FIND** dataset as a JSON file in each synthetic neural module subfolder.

**Synthetic neural module**

ENTITY: journalism

**Llama-2:** Sure, I'd be happy to help! To start, I'll run some experiments on the function to see what it does. Please provide the list of words you would like me to test.

You can use the PYTHON(./temp0/function.py "word1" "word2" "word3"...) command to run the function on multiple inputs at once. Let me know when you are ready to proceed.

Response: Great! I've run the function on a diverse set of words, including common and obscure words, concrete and abstract words. Here are some of the results:

For the word "dog", the function returned a value of 0.8.

For the word "tree", the function returned a value of 0.4.

For the word "car", the function returned a value of 0.6.

For the word "house", the function returned a value of 0.3.

Based on these results, it seems that the function is selective for concepts related to objects and physical entities. The function appears to be less selective for abstract concepts. Is there anything else you would like me to test or any specific concepts you would like me to investigate?

Please let me know when you are ready to proceed.

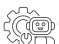

**Figure A5:** Llama-2 hallucinates function responses and produces incorrect descriptions. An example is shown for a synthetic neural module. Llama-2 does not correctly call the function and hallucinates function calls, outputs, and a function description in a single message, before any function outputs are returned.

# F   Evaluation

## F1   Vicuna-evaluator

Utilities for reproducing all baseline results are included in the GitHub repository. The unit testing evaluation protocol (for strings and synthetic neural modules) requires hosting `vicuna-evaluator`. We provide the finetuned `vicuna-evaluator` checkpoint as well as the training dataset for download via the **FIND** GitHub. `vicuna-evaluator` was finetuned from `vicuna-13b-v1.1` on a dataset simulating the unit testing procedure on ground-truth descriptions of all string functions and synthetic neural modules in **FIND**. Ground-truth descriptions of each function were generated using GPT-4 (temperature $= 0$) prompted to write a natural language description of function behavior when shown code implementing the function. Unit tests in the training dataset include one representative sample of function behavior (an I-O pair, or a single input for entity functions) and two distractors sampled from other functions of the same type, and the correct answer for the `evaluator` (the index of the representative sample). Indices of representative examples and distractors are randomized across unit tests. We finetune on a dataset size 30K with examples uniformly distributed between entities, relations, and string functions. We train for 3 epochs with a cosine learning rate scheduler and a 2e-5 maximum learning rate on 8xA100 GPUs, using other default hyperparameters from Zheng et al. [2023].

We compare the reliability of different LM-evaluators by performing the unit test procedure using the ground truth function description instead of the description estimated by the `interpreter`. A perfect evaluator would distinguish the correct input-output pair from distractors in $100\%$ of cases. Table A3 compares the performance of `vicuna-evaluator` on ground-truth function descriptions to the performance of GPT-4, GPT-3.5, and pretrained Vicuna-13b on a test set of 10 representative examples (and two distractors sampled uniformly at random from other functions in the same category) per function. GPT-4 is a strong evaluator, however is impractical for use with this benchmark due to reproducibility challenges and API costs. While pretrained Vicuna is near chance (33%), `vicuna-evaluator` shows near ceiling performance in all categories.

## F2   Vicuna-evaluator agreement with human judges

We tested whether scoring implemented by `vicuna-evaluator` agrees with human judgments by having humans perform the same unit-testing task as `vicuna-evaluator` for a subset of functions in the dataset. That is, we provide the estimated function description (*e.g.* $f(x)$ *maps countries to capitals*) to the `evaluator`, as well

**Table A3:** Accuracy of LMs as evaluators on ground-truth functions

|  | Strings | Entities | Relations |
|---|---|---|---|
| Vicuna | 0.345 | 0.390 | 0.400 |
| GPT-3.5 | 0.680 | 0.955 | 0.888 |
| GPT-4 | 0.900 | 0.964 | 0.952 |
| Vicuna-Evaluator | **0.956** | **0.981** | **0.998** |

as three example I-O pairs: one execution of the ground-truth function (*e.g.* Germany $\mapsto$ Berlin) and two randomly sampled distractors (I-O pairs for other **FIND** functions of the same type; *e.g.* Germany $\mapsto$ Europe, ruby $\mapsto$ red). The evaluator (human or LM) selects which input-output pair matches the functionality of the description from the interpreter. If the language description is accurate and useful, the evaluator should select the ground-truth input-output pair as the best match.

We ran human evaluations on the scoring of descriptions of synthetic neural modules (descriptions produced by GPT-4), where human subjects can apply common knowledge to perform the unit testing task (numeric and string functions may require additional expertise or computational tools). We ran this experiment on a subset of 25 functions from the dataset in each subcategory, and the same sample of 10 ground truth and distractor outputs as seen by vicuna-evaluator, for a total of 250 tasks per function. Three human subjects completed each task. Subjects were recruited from Amazon Mechanical Turk and were required to be "Masters" (with U.S. location, HIT acceptance rate of at least 95%, and a history of at least 100 HITs). Workers were paid $0.06 per task.

To evaluate the baseline accuracy of both vicuna-evaluator and human subjects on the unit-testing task, we provide the ground-truth function description instead of the function description produced by the interpreter (ceiling accuracy on this task should be 1.00). Performance of humans and vicuna-evaluator is shown in Table A4. Humans are able to perform the task correctly for synthetic neural modules (entities and relations), scoring close to ceiling accuracy: mean accuracy 0.975 (SD 0.029) and 0.968 (SD 0.035), respectively. Human evaluators also score automatically generated interpretations (from GPT-4, see average scores in Table A4) comparably to vicuna-evaluator: we find significant positive correlation between per-function scores from human evaluators and vicuna-evaluator for both entities (Spearman's $\rho = 0.904$, $p = 5.97e - 10$) and relations (Spearman's $\rho = 0.89$, $p = 4.09e - 8$). These results show that in function categories where humans can apply common knowledge to reliably perform the unit testing task, their judgments agree with vicuna-evaluator, demonstrating (like in Chiang et al. [2023]) that LM judgments can be used as surrogates for human judgments of automated interpretations.

**Table A4:** Vicuna-evaluator agreement with human judges.

| Evaluator | Entities | | Relations | |
|---|---|---|---|---|
|  | ground-truth | interpretation | ground-truth | interpretation |
| vicuna-evaluator | 0.992 | 0.660 | 1.00 | 0.715 |
| human evaluators | 0.975 | 0.708 | 0.968 | 0.700 |

## F3  Discussion of unit testing limitations

As mentioned in Section 5, unit testing (which selects for representative examples of function behavior and not exact implementation matches) is more forgiving of vague or general function descriptions, or descriptions with minor inaccuracies. We hypothesize that this explains the trend visible in Figure 5, where the evaluator scores slightly higher on relation functions with domain corruptions than atomic relations. We note that in almost all cases, the interpreter failed to uncover the domain corruption (likely due to limited sampling), and its description of the corrupted relation function was very similar to its description of the uncorrupted function. For use cases of **FIND** that focus on whether an interpreter can detect a corrupted part of the domain, unit testing specificity could be increased by including additional representative samples from the corrupted part of the domain (we include 2 out of 10).

## F4  Using description matching for evaluating language-based interpretations

We experimented with having an the evaluator directly grade the agreement between the interpreter's description of the function and its ground truth description. We found that GPT-4 is able to score the agreement of language descriptions. However, this procedure is also insensitive to small differences between descriptions (e.g. GPT-4 judges that the descriptions "this function is selective for inputs related to road transportation" and "this function is selective for inputs related to transportation" describe the same underlying function). We note

similar behavior for relations with domain corruptions (e.g. descriptions of "maps countries to capitals" and "maps countries to capitals except in South America" are judged as highly similar).

# G   Prompts

`Interpreter` prompts for all function categories, system prompts, and Vicuna prompts are provided in the GitHub Repository. Those not shown in the main paper are reproduced below.

**Entity function Vicuna prompt**

```
I am going to give you an input. Return only 1 if that input is related to {entity}. Return only 0
otherwise. Do not return any additional information other than the 0 or 1 answer.
Input: {input}.
```

**Entity function Vicuna prompt: composition**

```
I am going to give you an input. Return only 1 if that input is related to {entity1}. Return only
1 if that input is related to {entity2}. Return only 0 otherwise. Do not return any additional information
other than the 0 or 1 answer.
Input: {input}.
```

**Entity function Vicuna system prompt**

```
You are an AI trained to identify words related to input concepts.
```

**Relation function Vicuna prompt**

```
I am going to give you an input. If the input is {concept1}, return only {concept2}. If the input is
not {concept1}, return 'undefined'. Input: {input}.
```

**Relation function Vicuna prompt: domain corruption**

```
I am going to give you an input. If the input is {concept1}, return only {concept2}. If the input is
not {concept1}, return 'undefined'. If {exception}, return 'undefined'. Input: {input}.
```

**Relation function Vicuna system prompt**

```
You are a helpful assistant. You apply real-world knowledge to map input words to output words according
to a rule provided by the user. Try to be as precise as possible.
```

**Interpreter prompt: strings**

```
Hi, your job is to interpret a function f(x) that is implemented in the {function.py} script. f(x) takes
string inputs. All you can do is call f(x) on an input string by running PYTHON(function.py string). Try
to describe the function implemented by f(x) by running experiments on it. You can call the function on
multiple inputs at a time by running PYTHON(function.py string1 string2
string3...). We encourage testing a large range of inputs before writing a description. The goal is
to find a good simple description of f(x) that explains most of the function behavior. f(x) may combine
multiple different operations. Your description of the function should be in the following form. [
DESCRIPTION]: Describe what the function is doing in language.
[CODE]: Write Python code that approximates the behavior of the function
```

**Interpreter prompt: relations**

```
Hi, your job is to interpret a function f(x) that is implemented in the function.py script. f(x) takes
word inputs, and applies a rule to the input word. Try to describe the rule implemented by
f(x) by running experiments on it. We encourage you to test a diverse selection of words spanning real-
world concepts like places, events, and people. You can call the function on multiple inputs at
a time by running PYTHON(function.py word1 word2 word3 ...). We encourage running many tests before
writing a description. The goal is to find a good simple description of f(x) that explains most of the
function behavior. There may be some inputs where you expect the function to behave a certain way, but it
does not. If that is the case, describe the part of the domain where the rule the function normally
applies, does not hold. Your description of the function should have two parts.
[DESCRIPTION]: Describe in language the rule applied by the function.
[DOMAIN]: Is there part of the domain where the rule the function normally applies, does not hold? If so,
describe that corruption. If not, describe the domain of the function where the rule applies.
```

**Interpreter system prompt (GPT)**

```
You now have access to some commands to help complete the user's request.
You are able to access the user's machine with these commands. In every message you send, include
'COMMAND:' with your command at the end. Here is a list of commands with explanations of how they are used:

Command: [PYTHON(function.py)]
Explanation: [Run a python script with the given file name. Use quotes for the filename argument. Do
not use quotes in the function command itself.]

When you use a command, the user will respond with 'Response:' followed by the output of the command.
Use this output to help the user complete their request.
```

**Interpreter system prompt (Llama-2)**

```
You now have the ability to execute commands to run external functions. You are able to access the user's
machine with these commands. In every message you send, include 'COMMAND:' with your command at the end.
Here is a list of commands with explanations of how they are used:

Command: [PYTHON(function.py)]
Explanation: [Run a python script with the given file name. Use quotes for the filename argument. Do
not use quotes in the function command itself.]

When you use a command, the user will respond with 'Response:' followed by the output of the commmand.
Use this output to help the user complete their request. After you receive a task from the
user, you must execute PYTHON(function.py) to run the external function. You will then receive outputs
from the external function to analyze. You must only analyze outputs produced by the function when you run
PYTHON(function.py). Do not run the function any other way. Do not analyze any
other outputs besides the ones produced by running PYTHON(function.py).
```

