# OpenReview forum: "FIND: A Function Description Benchmark for Evaluating Interpretability Methods"
_NeurIPS.cc/2023/Track/Datasets_and_Benchmarks — NeurIPS 2023 Datasets and Benchmarks Poster_

### Official Review · Reviewer_riiN · 2023-07-18
**A framework for evaluating the reasoning capabilities of LLMs**

**Rating:** 8
**Confidence:** 3
**Correctness:** The framework appears to be correctly…
**Clarity:** The paper is very easy to read and fo…

**Strengths:**

+New framework for explaining black-box models using LLMs
+I have not seen any similar papers published

**Additional Feedback:**

This is a well written paper for evaluating the capabilities of LLMs to explain the behavior or black box models. This appears to be a new line of XAI research.  The reviewer is overall positive to this submission.

**Documentation:**

The framework appears to be well documented.

**Ethics:**

No problems.

**Limitations:**

The authors have discussed limitations such as poor search strategies.

**Opportunities For Improvement:**

- The prompting to the LLM requires a very precise description.

**Relation To Prior Work:**

The paper reviews related works and discusses the relation of FIND with respect to these work. There are no similar studies in the literature.

**Summary And Contributions:**

This paper proposes a framework called FIND to evaluate the capability of LLM to provide an transparent explanation for a black-box model.
The framework focuses on four different types of black box models, numeric, strings, entities, and relations. The atomic functions are used to  create complex functions through composition, corruption, approximation, and noise. The LLM is given the input and output format of the black-box model and is instructed to repeatedly query the model to understand its behavior. When the LLM believes that it understands the model, it provides an domain-specific explanation defined by the input prompt. For numeric functions, this includes a text interpretation, domain description, and code. The paper also outlines a few metrics for evaluating the explanations. The MSE error is for example used for numeric functions.

The main contribution of this framework is the ability to ask LLM to provide explanations for black-box models. This can be used to understand complicated systems and evaluate the reasoning capabilities of LLMs. This reviewer enjoyed reading this paper.

---

> ### Author Response · Authors · 2023-08-11
> **Response to Reviewer riiN**
>
> **A. The prompting to the LM requires a very precise description.**
> That’s true. We find the interpreter to work more precisely when given a detailed prompt. While this is the case for current models, we anticipate future models might show better performance interpreting functions even with more concise prompts.

---

> > ### Comment · Reviewer_riiN · 2023-08-29
> > **LLM prompt**
> >
> > I am satisfied with the authors response. I have also checked the concerns raised by the other reviewers and I am satisfied with the responses provided by the authors. I appreciate the effort of including a few more open source LLMs in the evaluation. I am also curious how the framework would work for more complex functions. However, that would be outside the scope of this paper.

---

### Official Review · Reviewer_FzK6 · 2023-07-20
**A crazy, bold and groundbreaking idea, but not yet realized**

**Rating:** 4
**Confidence:** 3

**Strengths:**

Ambitious problem.

A very solidly developed collection of diverse and complex functions describing a variety of problems.

A well-prepared framework to explore these functions (including an interpreter using a different language model - vicuna).

Solidly analyzed ability of GPT 3.5 and GPT 4 in exploring and explaining functions from the benchmark.

**Additional Feedback:**

On the technical side, a very interesting benchmark, extensive and interesting.
However, I find it difficult to agree with the conclusions, especially since the appropriate baseline for the results was not specified.

**Clarity:**

The motivation could be better outlined.
The rest of the article is very readable, well-documented and extensive.

**Correctness:**

Leaving aside overstated assessments of LLM capabilities, the rest of the conclusions and results are solidly presented.

Some disadvantage of the discussed solution is that one language model evaluates the correctness of another language model. It is not clear what the consequences of such model verification are.

**Documentation:**

Very well developed documentation and examples.
Extensive descriptions both in the article and in the examples on GitHub.

**Ethics:**

I don't see any immediate issues

**Limitations:**

The authors overestimate the obtained results of LLM evaluation. The section on limitations is quite short and the paper ends with the sentence "Baseline results show early evidence that advanced LLMs can construct scientific theories and execute experiments to validate them, supporting their suitability as a general-purpose interpretability backbone." with which it is difficult to agree.


**Opportunities For Improvement:**

The current version of the work has two significant weaknesses.

1. baseline. Of course, one can always calculate MSEs and compare them between GPT 4 and GPT 3.5 and one of them will always come out better, but what does that tell us about how far we are from, say, human experts?
Looking at the sample explanations in the https://github.com/multimodal-interpretability/FIND/blob/master/src/notebooks/example_interpretations.ipynb file, one can conclude that gpt-4 understands very little of the analyzed functions and its predictions are very far from ground truth.
In order for this benchmark to be useful, it should be used to compare either a larger number of models or refer to the effectiveness of humans.

2 If the goal is to see how LLMs can explain the discovered functions then the evaluations should be more complex than by MSE alone.

**Relation To Prior Work:**

Sufficient

**Summary And Contributions:**

The purpose of the paper is to evaluate to what extent language models can plan experiments that allow scheduling tasks to understand and describe how a given function works.

The paper presents a benchmark consisting of more than 2,000 functions (numeric, logical, textual) and an interactive process that allows exploration of the behavior of these functions.
This collection of tasks was prepared to evaluate the potential of language models in planning experiments and , "guessing" the behavior of the explored function.

This goal is very ambitious. The authors to heat up the importance of the results further state that:
"We show that an off-the-shelf LLM, augmented with only the capability to run an opaque function, can perform this task and act as a scientist, forming hypotheses, proposing experiments, and updating hypotheses in light of new data, even for programs involving composition of functions."

This statement, however, is difficult to agree with. While the benchmark itself is of great value, its indispensable element is a good baseline, defining a reference point for the measured results.
In the scenario at hand, the best benchmark would be human performance in exploring and explaining function performance.

---

> ### Author Response · Authors · 2023-08-11
> **Response to Reviewer FzK6 (1 of 2)**
>
> **A. For interpreting the results, a larger number of models are needed, or refer to the effectiveness of humans.**
> Great points. Following this comment we did both:
>
> (1) Benchmark additional LMs: We added Llama-2-13b and Llama-2-70b to our benchmark, performance is reported in the general comment above. Both Llama-2 models show inferior interpretation abilities compared to the tested GPT models.
>
> (2) Validate our evaluation protocol: we measure the agreement between our LM evaluator and human rankers on the unit-testing protocol and find high correlation between the two (see general comment above). We thus conclude the LM evaluator is a suitable surrogate for human judgments of function interpretation.
>
> **B. Evaluations should be more complex than by MSE alone.**
> Code-based (MSE) evaluations are a small part of our evaluation framework. We also describe in Section 4 a unit-testing protocol for evaluating language descriptions of functions. In this procedure, an LM evaluator selects which of three function input-output pairs corresponds to a language description of a function. To judge the interpreter’s accuracy, we provide the estimated function description (*e.g. f(x) maps countries to capitals*) to the evaluator, as well as three example input-output pairs: one execution of the ground-truth function (e.g. *Germany* → *Berlin*) and two randomly sampled distractors (input-output paris for other functions of the same type, e.g. *Germany* → *Europe*, *Ruby* → *Red*). The evaluator selects which input-output pair matches the functionality of the description from the interpreter. If the language description is accurate, the evaluator should select the ground-truth input-output pair as the best match.
>
> As described in the general comment, we finetuned a custom evaluator-LM with a vicuna backbone (vicuna-evaluator) to perform this unit testing task, and validate that its judgments align with human judgments of the same function descriptions.
>
> **C. The authors overestimate the obtained results of LM evaluation**
> We disagree. We found that off-the-shelf LMs demonstrated sophisticated experimentation strategies: making hypotheses based on observed behavior, and generating inputs that specifically test those hypotheses (see an example in Figure 3). We have added additional examples to the Appendix. In many cases, interpreters managed to reveal very interesting phenomena. For example, GPT-4 was able to successfully describe some synthetic neural modules representing logical compositions of concepts:
>
> **Ground truth entity function**
> I am going to give you an input. Return only 1 if that input is related to 'online communities'. Return only 1 if that input is related to 'music genres'. Return only 0 otherwise. Do not return any additional information other than the 0 or 1 answer.
>
> **Estimated description**
> GPT-4: The function appears to be selective for concepts related to music genres and online platforms. High values are associated with music genres like hip-hop, jazz, rap, and rock, as well as online platforms like Reddit, Twitter, and YouTube. The function assigns lower values to other categories such as sports, politics, food, and cars.
>
> In another example, GPT-4 recovered a non-trivial relation:
>
>
> **Ground truth relation function**
> I am going to give you an input. If the input is a country, reply with only another country on its border. If the input is not a country, return 'undefined'. Reply with a single word. Do not include any additional information other than the output word.
>
> **Estimated description**
> For the input words representing locations, the function outputs a nearby or related location. For other types of input words, the function returns “undefined.”

---

> > ### Author Response · Authors · 2023-08-11
> > **Response to Reviewer FzK6 (2 of 2)**
> >
> > **D. What are the consequences of using a language model to evaluate the correctness of another language model?**
> > Thanks for raising this important point. To validate our evaluation protocol we did the following (both are described in detail in the general comment above):
> >
> > (1) Trained a custom LM-evaluator (based on vicuna-13b) dedicated specifically to our unit-testing task. We show that this evaluator achieves high accuracy on ground truth descriptions of a validation set.
> >
> > (2) Validated the agreement between our vicuna-evaluator model and human raters, and found a very strong correlation, indicating that our evaluator can serve as surrogate for human judgments (see general comment to all reviewers for analysis).
> >
> > **E. The motivation could be better outlined. The rest of the article is very readable, well-documented and extensive.**
> > Thank you for pointing this out. We will make an effort to better motivate the paper based on the general motivation described in the first answer in the general comments. In a nutshell, the functions in FIND are derived from past findings about specific phenomena observed in the wild, and thus exercise the kinds of interpretability behaviors that will be required for understanding real neural networks.

---

> > > ### Author Response · Authors · 2023-08-24
> > >
> > > We think the main objection given by FzK6 is that our claim regarding the success of LM interpreters is overstated. We've scaled back the claim to better distinguish between sophisticated experimentation strategies (with no additional training, interpreters appear to design tests that isolate particular aspects of function behavior) and the accuracy of final descriptions (some functions are still mislabeled). In response to the main opportunity for improvement, we have benchmarked additional models and shown our scoring protocol greatly aligns with human judgments (see the response to all reviewers). We think this addresses the key concerns and would love to hear from the reviewer if this is what they had in mind.

---

> > > > ### Comment · Reviewer_FzK6 · 2023-08-24
> > > >
> > > > My main critique was the benchmark/baseline performance. In the first version it was a comparison of GPT 3.5 to GPT 4. Adding the comparison to LLaMA improved this situation.
> > > >
> > > > Still, the biggest weakness of the presented results is the evaluation.
> > > > In future studies, it would be good to define reference points, i.e. at which results we can consider the exploration problem solved (e.g. what performance would a human expert have).
> > > >
> > > > Despite this critique, I think the article is a valuable contribution and I am closer to recommending it to be presented at the conference.

---

> > > > > ### Author Response · Authors · 2023-08-26
> > > > >
> > > > > Great, thank you for the feedback. Our evaluation procedure considers an interpretability problem "solved" when an explanation matches the ground truth function description. We also expect solutions produced by automated interpretation procedures to be legible and useful to humans. We thus introduced an automated scoring procedure that compares both code-based and language-based explanations of functions to ground-truth function descriptions, and validated that the accuracy judgments of our automated scorer align with the judgements of human subjects (see comment to all reviewers for experiment details).
> > > > >
> > > > > We don't expect humans to be able to manually solve all of the interpretability problems in the benchmark, as some require expertise beyond common knowledge (e.g. estimating black-box math functions).
> > > > >
> > > > > Please let us know if you have a different type of evaluation in mind and we would be happy to include it.

---

### Official Review · Reviewer_gaMy · 2023-07-21
**A good benchmark to evaluate LLMs**

**Rating:** 7
**Confidence:** 4
**Correctness:** Yes
**Clarity:** Yes, it is well written.

**Strengths:**

1. This paper addresses an important problem in interpretability research, i.e., lack of ground truth explanations for evaluating interpretation methods. The new benchmark dataset fills an important gap.
2. The paper is overall well written and I enjoy reading it.
3. The research appears technically strong. The functions are systematically generated reflecting real-world complexity. The LLM baseline experiments demonstrate initial capabilities.
4. The code, data, and experimental details are open sourced to enable reproducibility. This allows the benchmark to be built upon.

**Additional Feedback:**

None

**Documentation:**

Yes, there is sufficient detail on data collection and organization, etc.

**Limitations:**

Yes, the authors adequately addressed the limitations.

**Opportunities For Improvement:**

1. Currently, the authors only implemented two interpreter, i.e., GPT-3.5 and GPT-4. It would be great if more LLMs are evaluated, such as Vicuna and Alpaca.
2. The complexity and diversity of functions represented could be expanded further to better reflect real-world AI systems.


**Relation To Prior Work:**

The paper does a good job distinguishing this contribution from previous work.

**Summary And Contributions:**

Recent advances in large language models (LLMs) suggest they could provide a backbone for interpretability tasks and an effective interface for end-users, however current evaluation methods for new explanation systems are limited and ad-hoc with no scalable datasets with ground-truth explanations to evaluate global interpretations. The paper presents FIND, a new dataset with over 2000 functions and corresponding explanations across textual/numeric domains where explanations represent real-world complexity like noise, compositions, approximations, and biases. Experiments provide an LLM-based interpretation baseline that constructs explanations in multiple forms - language, code, and structured Q&A, showing LLMs can act as scientists, forming/updating hypotheses and proposing experiments even for composed functions, indicating LLMs are an effective starting point for interpreting systems across diverse domains.

---

> ### Author Response · Authors · 2023-08-11
> **Response to Reviewer gaMy**
>
>
> **A. Evaluate more LMs**
> We agree that benchmarking more models is important. Following this comment we added Llama-13b-chat and Llama-70b-chat to our benchmark. Performance is reported in the general comment above.
>
> **B. Functions could better reflect real-world AI systems**
> Our dataset is constructed to represent relevant interpretation scenarios in multiple domains. Please see our response regarding this point in the general comment. We will better highlight this motivation in the revised version of the paper.

---

### Official Review · Reviewer_C5TS · 2023-07-22
**A Well-Established Function Interpretation Benchmark with Advanced LLMs**

**Rating:** 7
**Confidence:** 3
**Correctness:** The dataset is constructed in a reaso…

**Strengths:**

1. Function interpretation is a very crucial task, especially in the era of large language models, because researchers still don't understand how advanced large models perform these tasks.

2. This work brings the new dataset and benchmark to the eXplainable AI research community.  Given that there are currently no scalable datasets across diverse domains that provide ground-truth explanations for evaluating global interpretations of structured functions, the well-designed dataset in this work may have a significant impact on XAI.

3. There is a debate in the research community about whether it is reasonable to use large models to measure response quality. This work shows early evidence that advanced LLMs can construct scientific theories and execute experiments to validate the interpretation of functions.

**Additional Feedback:**

None.

**Clarity:**

This paper is well-written and easy to understand. The presentation of Section 3 can be improved by better positioning the prompt examples.

**Documentation:**

Delivered in a good quality.

**Ethics:**

No.

**Limitations:**

1. Instead of proposing a new LLM, this work measures the off-the-shelf advanced LLMs' ability for supporting their suitability as a general-purpose interpretability backbone.

2. Following the last point, using off-the-shelf LLMs as interpreters may miss small corruptions to part of the domain, which in real-world interpretability settings could stem from bias in the training set, also mentioned by authors in the paper.

3. If my understanding is correct, FIND is limited to providing evaluations on functions whose ground truth structure is known. I understand that employing crowd-sourcing works to conduct subjective evaluations for questions that do not have ground truth is not the main focus of this paper, while the subjective evaluation of explanation quality has been ignored in this work, which is also important towards the XAI research community.

**Opportunities For Improvement:**

Some important details of how FIND API works in missing from the current version. Authors could consider adding the description of FIND API to Appendix.

**Relation To Prior Work:**

Yes.

**Summary And Contributions:**

This work presents FIND: a Function INterpretation Dataset and benchmark for evaluating interpretability methods on functions whose structure is known a priori based on existing work [1]. The dataset consists of more than 2,000 functions and corresponding explanations that are procedurally constructed across textual and numeric domains, which have been open-sourced and are available on GitHub. The FIND dataset is built from a diverse set of programs including explicit mathematical equations, string operations, logical composition, lexical relations, and factual associations. After processing datasets with some complex functions, including compositions, biases, approximations and noise, the input data will be fed into an interpreter, referring to GPT-4 in this work. The benchmark comes
with an initial evaluation framework that measures an interpreter’s ability to predict the outputs of a processed target function. Finally, they provide an evaluation by comparing text and code interpretations to ground-truth function implementations.


[1]. B. Z. Li, J. Yu, M. Khabsa, L. Zettlemoyer, A. Halevy, and J. Andreas. Quantifying adaptability in pre-trained language models with 500 tasks. arXiv preprint arXiv:2112.03204, 2021.

---

> ### Author Response · Authors · 2023-08-11
> **Response to Reviewer C5TS**
>
> **A. Details about the FIND API are missing.**
> The FIND API is described in the Appendix. Our API includes a function generation module that generated the numeric and string functions in FIND, an interpretation module that performs an iterative interpretation process by interacting with LLMs, and an evaluation module that scores interpretation according to the indicators described in the paper. Please find additional details and examples for functions in Appendix A1.
>
> **B. Paper benchmarks off-the-shelf LMs instead of proposing a new LM; these models may miss small corruptions to part of the domain**
> True! The primary contribution of this work is to propose a benchmark suite for evaluating the efficacy of different interpretation systems at evaluating functions that resemble sub-computations inside larger trained neural networks, not to propose a new LM or interpretation technique. While the task of training a LM to do interpretation is interesting by itself, and may improve performance detecting noise and corruptions, we leave it to future work.
>
> **C. Subjective evaluation of explanation quality (for questions that do not have ground truth) is outside the scope of this paper, but important to XAI.**
> The main motivation of our work is to benchmark interpreters on functions with ground-truth explanations that mimic functional behaviors in real-world interpretability settings (see motivation in the general comment above). Explanations produced by automated interpretability procedures have led to subjectively meaningful outcomes when derived for real models (e.g. the model-editing experiment in [1] uses automatically-constructed neuron labels for intervention on a model that improves performance in an adversarial test setting)
>
> [1] Hernandez et al. "Natural language descriptions of deep visual features."

---

> > ### Comment · Reviewer_C5TS · 2023-08-24
> >
> > The response addressed all my concerns, I am happy to recommend this paper to be presented at the conference.

---

### Official Review · Reviewer_fw4P · 2023-07-22

**Rating:** 6
**Confidence:** 4
**Clarity:** Yes

**Strengths:**

Interpretability benchmarks are sorely needed and I am very happy to see work in this direction! Interpretability is an extremely important research area that has been hampered by the lack of 1) tractable yet meaningful goals to drive the field towards, and 2) the ability to confidently measure progress towards those goals

The benchmark gets points for making a serious and thoughtful attempt at (1) and (2) including a variety of kinds of functions to interpret, and (with a few caveats described in the next section) well-crafted evaluation of code and text interpretations.


**Additional Feedback:**

Please take my suggestions above as constructive—I am very eager to see this line of work succeed, and would be willing to increase my score based on substantive changes to the paper (especially interpretation tasks that are grounded in the complexities exhibited by real AI systems)

**Correctness:**

I didn’t look at the code, but the methods seemed reasonable modulo concerns above.

**Documentation:**

Yes, module concerns about baselines above

**Limitations:**

I found the limitations section to be quite thin and would like to see much more discussion of aspects discussed above.

**Opportunities For Improvement:**

The importance of this direction is extremely high, but unfortunately it’s a very challenging direction to operationalize well. Here are some opportunities I see for improvement in this attempt:

## Ecological validity

The biggest opportunity for improvement I see is with respect to ecological validity. This describes how well progress on a benchmark will translate to progress towards the end goal we really care about: understanding AI systems. Put in the most simple of terms, I’m not as convinced as I’d like that methods that perform well on FIND would help us with understanding AI systems better.

The most compelling way I could see FIND helping is in improving tasks like neuron interpretation. Neurons can fire on a range of different stimuli, including compositions of unrelated stimuli, and this maps nicely to the Text Modules part of the benchmark (for language models) and the composition operators presented in the paper. This connection would benefit from being drawn out more strongly.

However, I’m not as convinced by many of the numeric functions or string functions in the benchmark. These are probably the first kinds of functions I would pick if I were trying to think of functions I could evaluate easily. But I’m not convinced that these functions are characteristic of the kinds of functions we expect to find inside of neural networks (whether for networks trained on natural language, programming languages, medical images, weather patterns etc) or would help us interpret those networks better. I think the paper could be improved by better motivation here, better acknowledgment of limitations, and (most importantly) a more creative and realistic choice of tasks that engages with this challenge.

Furthermore, another important dimension in interpretability is neural circuits: we don’t just want to know what the neurons in our networks do, but we also want to know how they fit together. It may not sufficient to do composition of random, unrelated functions here—interpretability needs to discover the roles that different parts of a network play towards a larger purpose. Here, you could potentially look at previously-studied mechanisms inside neural networks, such as indirect object identification, modular arithmetic, etc. Of course, the benchmark doesn’t have to cover this behavior, but if it doesn't it would be helpful to characterize better how the benchmark might fit into the broader goals of the field.

What I wouldn’t want to see happen is for FIND to contribute to a streetlight effect for interpretability research, where as a field we choose to focus on simple functions that we can easily generate + evaluate as opposed to the harder but more realistic behaviors we'll need to understand in order to make progress understanding AI systems.

In sum, my assessment is that the benchmark attempts to help us understand AI systems better, but chooses to analyze simple functions instead of realistic AI systems. More work is needed to either justify this gap and / or close it.


## Methodological choices

1. It’s nice that the evaluation methods can actively probe the model, but I find the setup a bit strange. In general won’t we have a bunch of examples from the dataset that a neuron activated on, as in the setup from Bills et al 2023?
2. Reproducibility of the baseline. Perhaps more than in any other paper, reproducible baselines are crucial for benchmarks. Given the stated depreciation policy of the OpenAI API, I have deep reservations about having the only baselines be implemented with models that might not even be available by the time NeurIPS rolls around in December.
3. [Smaller point] It could be nice to acknowledge the limitations of MSE—for example, in 4(a) if the dip in the function was smaller the model could still ignore the dip entirely and keep the NMSE < 0.1
4. [Smaller point] It would be helpful to discuss the limitations of the distractor-based evaluation for the text descriptions. Should we worry that distractors might not catch finer distinctions (e.g. transport vs ground transport)? Did you consider having an LM grade how similar the text descriptions were?

[Bills et al 2023] Language models can explain neurons in language models


**Relation To Prior Work:**

I saw no discussion of the relation of this work to the program induction literature, which studies very similar questions. Probably that deserves a thorough literature review + paragraph in the related works section.

**Summary And Contributions:**

The paper presents a benchmark for function interpretation: given a black-box function, it quantitatively evaluates an explanation of that function in the form of a textual description or code implementation. This enables benchmarking and comparison of function interpretation methods across a wide range of functions.

---

> ### Author Response · Authors · 2023-08-11
> **Response to Reviewer fw4P (1 of 2)**
>
> **I. Ecological validity**
>
> We appreciate your feedback and agree that scoping FIND within the broader interpretability landscape is highly important. In the new draft of the paper, we have revised the introduction to (a) better characterize the toy problems that FIND currently captures as useful (but not comprehensive) building blocks of automated interpretability, and (b) distinguish what FIND does not yet include (circuit-based problems, networks trained on real-world data). The comment to all reviewers describes our thinking in more detail. We have updated Section 2 to better motivate the design decisions around individual function categories, and included discussion of the current scope of the benchmark and shortcomings of the evaluation protocol (see C, D below) in the revised limitations.
>
> **II. Methodological choices**
>
> **A. Experimental setup: aren’t maximally activating exemplars usually precomputed?**
>
> Yes, many interpretability settings involve precomputed sets of maximally activating exemplars. Our broader interactive setting leaves room for interpreters that also generate the data they describe. Importantly, however, methods that “dissect” FIND functions to precompute exemplar sets that automated interpretability techniques then describe are certainly not excluded from the benchmark. We include two experiments (one in the submitted paper, an additional in the revision) that simulate existing interpretability procedures that label precomputed exemplars, and evaluate the performance of these procedures on synthetic neural modules in FIND.
>
> The submitted version of the paper includes in Section 4 an experiment where off-the-shelf LMs are equipped with a fixed set of inputs to use for probing the functions during interpretation. These inputs were intended to directly mirror the maximally activating exemplars used by NetDissect [2], MILAN [1], and recent MILAN-type LM analysis methods. In this setting, interpreters can conduct additional tests after observing function outputs on the provided inputs. Unsurprisingly, this setting vastly improves interpreter performance (described in Table 2 - “w/ init”).
>
> We view exemplar computation as one of many “tools” that an interpreter could use, and hope that this benchmark will drive exploration of additional tools (e.g. DeepDream-style example synthesis) as part of automated interpretation methods.
>
> The revised paper includes a second experiment that more closely resembles procedures such as MILAN [1] that used learned models to describe maximally activating exemplar sets (where subsequent experimentation is not allowed). Performance of “GPT-MILAN” is weaker on synthetic neural modules than in our “initialization” setting that permits additional experimentation. This suggests that approaches that combine fixed exemplar sets with additional, interactive experimentation could potentially outperform many existing labeling procedures in two regards:
> (i) Efficiency:  techniques like Network Dissection [2] use huge fixed datasets as inputs for precomputing maximally activating exemplars. Potentially, a  much smaller set of dataset of test cases is necessary if additional experimentation by an interpretability model is possible.
> (ii) Flexibility: experimentation might surface novel behaviors not captured in predefined sets of exemplars. Interpreters could test new inputs in subspaces that might be not well covered in fixed datasets.
>
> We have updated Section 4 of the paper to include the additional GPT-MILAN experiment and an expanded discussion of benchmarking interpretability procedures based on description of precomputed exemplars.
>
> [1] Hernandez et al. "Natural language descriptions of deep visual features."
> [2] Bau et al. "Network dissection: Quantifying interpretability of deep visual representations."
>
> **B. Reproducibility of the baseline**
> Thanks for raising this point, we completely agree that the deprecation of GPT-4 poses a challenge for reproducing baseline results. To address this, we fine-tuned open-source Vicuna-13b to replace GPT-4 as evaluator, and will release the model checkpoint. See the general comment to all reviewers for more details.
>
> **C. Limitations of NMSE indicator**
> The NMSE indicator might be imperfect in capturing corruptions, nevertheless, results reported in Fig. 5 shows that the average performances reported by NMSE indicator for the corrupted numeric functions (as well as their noisy observations and MLP approximations) are inferior to that of the clean function, indicating that the difficulty of detecting corruption is reflected in the current evaluation procedure. We will add this discussion to the revised limitations.

---

> > ### Author Response · Authors · 2023-08-11
> > **Response to Reviewer fw4P (2 of 2)**
> >
> > **D. Limitations of distractor-based “unit testing” (did you consider having a LM grade how similar the text descriptions were?)**
> > This is an important point, thanks for highlighting it. We experimented with having an LM directly grade the agreement between the interpreter’s description of the function and its ground truth description. We found that an LM like GPT-4 is able to score the agreement of language descriptions. However, this LM is also insensitive to small differences between descriptions (e.g. GPT-4 judges that the descriptions “this function is selective for inputs related to road transportation” and “this function is selective for inputs related to transportation” describe the same underlying function). We note similar behavior for relations with domain corruptions (e.g. descriptions of “maps countries to capitals” and “maps countries to capitals except in South America” are judged as highly similar).
> >
> > The FIND unit-testing procedure enables selection of cases specific to function behaviors (e.g. the input “airplane” could distinguish between functions selective for road transportation and transportation more broadly). The sensitivity of unit-testing to the choice of distractors is also a potential limitation of the method. However, we test each function with 10 input-output triplets, reducing the likelihood of evaluation inaccuracies from distractors. We will add a detailed discussion of this issue to the revised limitations.

---

> ### Comment · Reviewer_fw4P · 2023-08-16
>
> Given the changes, I am happy to raise my score to a 6. I would increase further (to a 7 or even higher) if the authors broadened the diversity and creativity of functions in the benchmark as I view this as particularly important.

---

> > ### Author Response · Authors · 2023-08-28
> >
> > Thanks for your feedback. We agree, and are including additional synthetic neural modules that operate at the sentence level and implement syntactic operations like dependency relations and identifying parts of speech. Such behaviors have been observed in the wild in transformer attention heads, for example in [1], [2], [3].
> >
> > Simple implementations of dependency relations as black-box functions test an interpreter's ability to decode operations relevant to transformer attention. These functions take an input $x$ of length $L$ and "focus attention" from $x_{L}$ to $x_i$ $(i \leq L)$, and return $x_i$ as the function output.
> >
> > For example, take $f(x)$ which focuses attention from the end of a noun phrase to the head noun.
> > $f$('The premise on which the United States of America is founded') $=$ premise
> > When $x$ does not include the pattern that the function targets, $f(x) =$ 'undefined'.
> >
> > Other examples include:
> > * previous token heads (attends to previous token position)
> > * induction heads (specifically, the second head in the induction circuit, which attends to the token following the previous instance of the present token $x_L$)
> > * duplicate token heads (identify tokens that have already appeared)
> > * null attention (directed to first token)
> > * heads targeting words preceding commas
> > * heads targeting negations
> >
> > Functions that identify parts of speech return all $x_i$ with a particular part-of-speech tag (e.g. noun, proper noun, verb, pronoun, conjunction...)
> >
> > We intend for FIND 2.0 to be a larger release that incorporates white-box interpretability tasks. Let us know if there are specific problems you would like to see included.
> >
> > [1] Vig, Jesse, and Yonatan Belinkov. "Analyzing the structure of attention in a transformer language model." (2019).
> > [2] Olsson, Catherine, et al. "In-context learning and induction heads." (2022).
> > [3] Wang, Kevin, et al. "Interpretability in the wild: a circuit for indirect object identification in gpt-2 small." (2022).

---

### Author Response · Authors · 2023-08-11
**General response to all reviewers**


We thank all reviewers for their very helpful feedback! Below we respond to comments that appeared in some form across multiple reviews, and address other comments in responses to individual reviewers.

---

> ### Author Response · Authors · 2023-08-11
> **Relevance of FIND functions to interpretability research**
>
> We view FIND as a test of necessary, but not sufficient, capabilities for automated interpretation. Obviously, the ultimate test of these interpretation methods’ effectiveness must be their ability to generate actionable insights about real models, which FIND does not evaluate. But clean, simple benchmarks with ground-truth answers have been a major driver of more general capabilities in LMs, and we hope that FIND can play a similar role in interpretability research. We intend for it to be a living benchmark, incorporating new functions, interaction paradigms, and evaluation metrics as interpretability methods grow in sophistication and improve our understanding of real-world model behavior correspondingly. Future versions of FIND will incorporate white-box function interpretation problems including descriptions of individual components of neural circuits (the IOI circuit from [1] is just one example), some of which could be represented as the types of composition problems that are already included in FIND, but where interpreters access and label individual sub-computations inside composed functions.
>
> As a starting point, FIND focuses on the black-box function description paradigm because black-box description appears to be a subroutine (or is the sole operation) implemented by almost every existing automated interpretation method, spanning label retrieval, program synthesis and learning-based approaches. The functions that make up FIND are derived from past findings about specific phenomena observed in the wild, including polysemanticity, compositionality, and mis-specification, and thus exercise the kinds of interpretability behaviors that will be required for understanding real neural networks. We have revised Section 2 of the paper to better motivate the design decisions around each functional behavior incorporated into FIND:
>
> * **Numeric functions** test a variety of capabilities needed to interpret basic mathematical operations in neural networks, which can include relations between neurons or a model’s representation of numeric inputs. An example use is in [2], where an algorithm using trigonometric manipulations was found to perform modular arithmetic in a toy transformer model.
> * **String functions** test a variety of string manipulations, which may be necessary for understanding parts of models that manipulate sequential inputs, such as attention heads in transformers. The simplest example of this is induction heads, which Olsson et al. [3] claim support in-context learning.
> * **Atomic entity functions** simulate individual neurons or subcomponents of trained networks that respond to specific classes of inputs. This phenomenon has been observed in language models, where Bills et al. [4] attempted automated interpretability of neurons in GPT-2, and in vision models, where Bau et al. [5] found many neurons selective for interpretable features.
> * **Composition of atomic entities** may help identify how subcomponents of trained neural networks perform logical composition, which has been observed in Fong et al. [6] and Mu et al. [7]. Polysemantic neurons (which respond to logical disjunctions of concepts) have been found by Gurnee et al. [8] in language models and Olah et al. [9] in vision models.
> * **Relation functions** are relevant to identifying stored relational knowledge in neural networks, such as in Meng et al. [10] where subcomponents of neural networks were found to implement information retrieval.
> * **Noise and input corruptions** are designed to eventually support targeted evaluation of interpreters’ ability to identify disparate model accuracy across input regions (relevant to fairness, accountability, and data poisoning attacks as in [11],[12]) and shortcut solutions to complex algorithmic problems (relevant to questions about reasoning and robustness).
>
> [1] Wang et al. "Interpretability in the wild: a circuit for indirect object identification in gpt-2 small."
> [2] Nanda et al. "Progress measures for grokking via mechanistic interpretability."
> [3] Olsson et al. "In-context learning and induction heads."
> [4] Bills et al. “Language models can explain neurons in language models.”
> [5] Bau et al. "Network dissection: Quantifying interpretability of deep visual representations."
> [6] Fong and Vedaldi. "Net2vec: Quantifying and explaining how concepts are encoded by filters in deep neural networks."
> [7] Mu and Andreas. "Compositional explanations of neurons."
> [8] Gurnee et al. “Finding Neurons in a Haystack: Case Studies with Sparse Probing.”
> [9] Olah et al. “Zoom in: An introduction to circuits.”
> [10] Meng et al. "Locating and editing factual associations in GPT."
> [11] Biggio et al. “Poisoning attacks against support vector machines.”
> [12] Steinhardt, Koh, and Liang. "Certified defenses for data poisoning attacks."

---

> > ### Author Response · Authors · 2023-08-11
> > **Benchmarking additional models (Llama2-13b-chat, 70b-chat)**
> >
> > Several reviewers (gaMy, FzK6) mentioned the value of benchmarking additional models.
> > We have added Llama-2-13b-chat and  Llama-2-70b-chat, and report benchmark results below (also added to Table 2 in the revised paper). We found that both Llama-2 instances often hallucinated the outputs of functions instead of interpreting outputs returned by the functions themselves. We have included examples of this hallucination behavior in the revised Appendix. We also attempted to use Vicuna-13b as an interpreter, but it displayed sufficient hallucination behavior that we didn’t consider it to be an informative addition to the benchmark. This is also discussed in the revised Appendix.
> >
> > |            | Code |     | Language |     |       |       |
> > |------------|------------|-----|----------------|-----|-------|-------|
> > |            | Numeric    | Strings | Entities |  | Relations |  |
> > | Indicator: | NMSE<0.1   | unit test | unit test | w/ init | unit test | w/ init |
> > | Llama-2-13b| 0          | 0.33     | 0.34    | 0.58  | 0.46  | 0.42  |
> > | Llama-2-70b| 0.01       | 0.33     | 0.34    | 0.62  | 0.47  | 0.46  |

---

> > > ### Author Response · Authors · 2023-08-11
> > > **LM-based evaluation of language descriptions of functions**
> > >
> > > In the submitted version of the paper, we employed GPT-4 as an evaluator of language descriptions of functions, using a unit-testing procedure where the evaluator distinguishes representative examples of function behavior from distractors.  Related work [1] has shown that LM-judges such as GPT-4 are scalable and accurate surrogates for human judgments, which are otherwise expensive to obtain. However, using such models to compare interpreter performance on a benchmark task poses reproducibility challenges as model versions are deprecated (as noted by Reviewer fw4p). For the revised paper, we finetuned an open-source LM evaluator based on vicuna-13b (“vicuna-evaluator”) to replace GPT-4. We will release the finetuned vicuna-evaluator checkpoint and training dataset, and provide training details in the revised Appendix. We show that when performing unit testing with ground-truth descriptions, vicuna-evaluator matches function descriptions to representative inputs and outputs on a test set more accurately than GPT-4, GPT-3.5, or pretrained Vicuna-13b (higher score is better):
> > > |                  | Strings | Entities | Relations |
> > > |------------------|---------|----------|-----------|
> > > | Vicuna           | 0.345   | 0.390    | 0.400     |
> > > | GPT-3.5          | 0.680   | 0.955    | 0.888     |
> > > | GPT-4            | 0.900   | 0.964    | 0.952     |
> > > | Vicuna-Evaluator | **0.956** | **0.981** | **0.998** |
> > >
> > > Several reviewers (C5TS, FzK6) also asked about human judgments of interpreter-generated descriptions. We tested whether scoring implemented by vicuna-evaluator agrees with human judgments by having humans perform the same unit-testing task as vicuna-evaluator for a subset of functions in the dataset. In the unit-testing task, we provide the estimated function description (e.g. f(x) maps countries to capitals) to the evaluator, as well as three example input-output pairs: one execution of the ground-truth function (e.g. Germany → Berlin) and two randomly sampled distractors (input-output paris for other functions of the same type, e.g. Germany → Europe, Ruby → Red). The evaluator selects which input-output pair matches the functionality of the description from the interpreter. If the language description is accurate and useful, the evaluator (human or LM) should select the ground-truth input-output pair as the best match.
> > >
> > >
> > > We ran human evaluations on the scoring of synthetic neural modules, where human subjects can apply common knowledge to perform the unit testing task (numeric and string functions may require additional expertise or computational tools).  We ran this experiment on a subset of 25 functions from the dataset in each subcategory, and the same sample of 10 ground truth and distractor outputs as seen by vicuna-evaluator, for a total of 250 tasks per function. Three human subjects completed each task. Subjects were recruited from Amazon Mechanical Turk and were required to be “Masters” (with HIT acceptance rate of at least 95% and a history of at least 100 HITs). Additional details about the experiment, worker qualifications, IRB approval, and human subject payment will be provided in the revised Appendix.
> > >
> > >
> > > To evaluate the baseline accuracy of both vicuna-evaluator and human subjects on the unit-testing task, we provide the ground-truth function description instead of the function description produced by the interpreter (ceiling accuracy on this task should be 1.00). Performance of humans and vicuna-evaluator is shown in the table below. Humans are able to perform the task correctly for synthetic neural modules (entities and relations), scoring close to ceiling accuracy: mean accuracy 0.975 (SD 0.029) and 0.968 (SD 0.035), respectively. Human evaluators also score automatically generated interpretations (from GPT-4, see average scores in table below) comparably to vicuna-evaluator: we find significant positive correlation between per-function scores from human evaluators and vicuna-evaluator for both entities (Spearman's ρ = 0.904, p = 5.97e-10) and relations (Spearman's ρ = 0.89, p = 4.09e-8).
> > >
> > > We show that in function categories where humans can apply common knowledge to reliably perform the unit testing task, their judgments agree with vicuna-evaluator, demonstrating (like in [1]) that LM judgments can be used as surrogates for human judgments of automated interpretations.
> > >
> > > |                   | entities-gt | entities-interp | relations-gt | relations-interp |
> > > |-------------------|-------------|-----------------|--------------|------------------|
> > > | vicuna-evaluator  | 0.992       | 0.660           | 1.00         | 0.715            |
> > > | human evaluators  | 0.975       | 0.708           | 0.968        | 0.700            |
> > >
> > >
> > >
> > > [1] Zheng et al. “Judging LLM-as-a-judge with MT-Bench and Chatbot Arena.”

---

### Decision · Program_Chairs · 2023-09-22

**Decision:**

Accept (Poster)

**Comment:**

The authors developed a novel framework they called FIND for explaining black-box models using LLMs. The LLM is given the input and output formats of the black-box model and is instructed to query the black-box model until it understand its behaviour. After than LLM can provide an explanation.
The paper presents a benchmark consisting of 2k+ functions, including numeric, logical, and textual. For numeric functions, the explanation includes a text interpretation, domain description, and code. Complex functions describing a great variety of problems are produced by means of composition, corruption, and approximation.
The reviewers appreciated the originality, ambition and potential of the proposed framework as well as the overall quality of the paper. The reviewers also raised valid critique and the authors addressed it adequately during the rebuttal phase. Notably, they better motivated the design decisions for each functional behaviour of FIND, and introduced new  experiments with open-source LLMs.  While there is room for further improving the benchmark and evaluation framework, the reviewers agree that the paper in its current form provide substantial contribution and can be accepted to the conference track.